# Learning Diverse Textual Contexts for Robust Personalization of Text-to-Image Diffusion Models

## Abstract

Text-to-image (T2I) personalization aims to adapt pre-trained T2I models based on user-provided example images for customized image generation. In existing personalization approaches, the models are typically trained with a small number of personal concept images captured in limited contexts. This often weakens robustness, resulting in poor alignment between the text prompts and the generated images. Existing approaches tackled this by collecting images, thereby introducing diversity in the personal concept's context. Despite its effectiveness, this is often impractical, considering the high cost of image collection. To circumvent this limitation, we instead diversify the contexts of the personal concept in *text space*. Based on the fact that the T2I personalization method represents personal concepts as text tokens *e.g., "[v]"*, this diversification can be easily achieved by composing the tokens with various contextual words *e.g., "[v] at Eiffel Tower"*, offering an efficient alternative to costly manual image collection. During personalization, we leverage these text prompts for training to learn diversified contexts. However, utilizing diversified text prompts for personalization is not straightforward, as T2I personalization typically requires paired images as learning targets. To achieve learning without requiring images, we propose to learn them within *text space*. Specifically, we leverage masked language modeling (MLM), which operates entirely within *text space*. By leveraging MLM during personalization, diversified contexts are learned without involving any images. We demonstrate the effectiveness of the proposed approach with extensive experimental results to show that diverse context learning with MLM yields notable improvements in prompt fidelity and state-of-the-art results on widely used public benchmarks. Furthermore, we present an analytical study showing how our approach influences representations in text space through cosine distance analysis of text embeddings, and how these effects propagate to image space via cross-attention maps analysis, providing evidence of its effectiveness.

## 1 Introduction

Diffusion-based generative models (Ho et al., 2020; Song et al., 2020a; Dhariwal & Nichol, 2021; Song et al., 2020b) have made significant progress in image synthesis, achieving improved diversity and expressiveness in generated outputs. Extending these breakthroughs, diffusion-based text-to-image (T2I) models (Rombach et al., 2022; Podell et al., 2023; Balaji et al., 2022; Saharia et al., 2022; Xue et al., 2024) trained on large-scale text-image pairs (Schuhmann et al., 2022) have demonstrated impressive capabilities in translating the text into visual content.

More recently, leveraging the strong prior knowledge of the pretrained T2I generative models (Rombach et al., 2022; Chen et al., 2024b), various approaches have been proposed to finetune the models for personalization (Gal et al., 2022; Ruiz et al., 2023; Kumari et al., 2023; Voynov et al., 2023; Avrahami et al., 2023). Typically, these methods leverage 4-5 images capturing personal concepts in limited contexts to fine-tune the T2I model for personalization (Figure 1, (a)). While demonstrating its potential, existing methods often suffer from *overfitting*, causing the customized model to generate images that are highly similar to the training images and fail to faithfully follow the text prompts during inference (Figure 1, (b)). Some approaches (Chen et al., 2024c; Wang et al., 2025) addressed

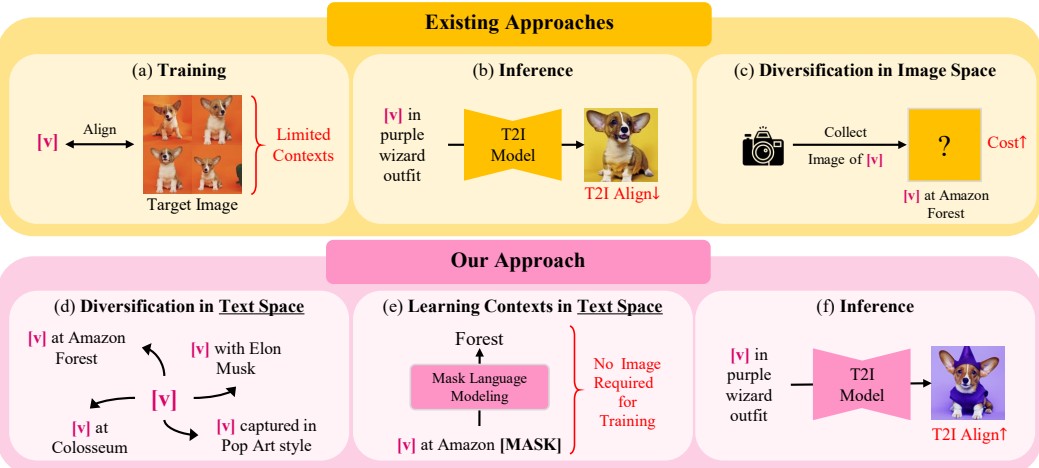

Figure 1: **An Illustrative Comparison Between Existing Approaches and Ours.** **(a)** In existing approaches, the model is finetuned to align the token *"[v]"* with target images collected under limited contexts. **(b)** This often causes overfitting to those limited contexts, resulting in poor text-image alignment. **(c)** Some approaches tackled this by collecting images captured in diverse contexts; however, this is often impractical due to the high cost of data acquisition. **(d)** Instead, we diversify the context of the personal concept within *text space*. Since the token *"[v]"* represents the personal concept, this can be easily done by composing the token with diverse context words. **(e)** We propose learning these diversified contexts in the *text space* by leveraging masked language modeling (MLM). While personalization typically requires paired images as learning targets, our approach circumvents this requirement, as MLM operates entirely within *text space*. **(f)** As the personal concept token learns diversified contexts, the model generalizes better to novel contexts during inference, leading to notable improvements in text-image alignment.

this by collecting images; however, this is often impractical, as images captured in diverse contexts are costly to obtain (Figure 1, (c)).

Instead, we propose to diversify the context solely within *text space*. Based on the fact that the personal concept token, *e.g., "[v]"* corresponds to the personal concept in T2I personalization (Gal et al., 2022; Kumari et al., 2023; Ruiz et al., 2023), this can be easily achieved by combining with diverse context words, *e.g., "[v] at Eiffel Tower"* (Figure 1, (d)). Utilizing diversified textual contexts, we propose learning them within *text space* using masked language modeling (MLM) (Figure 1, (e)). Unlike typical personalization, which requires paired images as learning targets, our approach circumvents this requirement, as MLM operates entirely within *text space*. By training the model to learn diverse contexts, the model generalizes more effectively to novel contexts during inference, resulting in notable improvements in text-image alignment (Figure 1, (f)).

Later, we show that adopting the MLM objective with a contextually diverse text prompt set during personalization alleviates the overfitting and leads to semantic enhancement in textual representation, which ultimately extends to higher prompt fidelity in image generation.

We conduct extensive experiments to demonstrate the effectiveness of our approach and outperform existing approaches. We summarize our contributions as follows,

- We introduce a novel T2I personalization method that enhances the contextual diversity of personal concepts within text space.
- We achieve learning diverse contexts *without requiring images*, by leveraging textual-space learning with a masked language modeling objective.
- We provide an analytical study demonstrating that our approach yields semantic enrichment in the textual representation of personal concepts, which in turn improves alignment and fidelity in the image space.
- We conduct extensive experiments validating the effectiveness of our method, showing consistent improvements and superior generation quality compared to state-of-the-art personalization approaches.

## 2 METHOD

### 2.1 PRELIMINARIES

**Self-Attention as Linear Combination.** We first would like to highlight the important aspect of attention layers: *the output token of the attention layer is the linear combination of the input tokens.* We elaborate this with the self-attention layer. Let $\mathbf{Q}_l \in \mathbb{R}^{L \times d}$, $\mathbf{K}_l \in \mathbb{R}^{L \times d}$, $\mathbf{V}_l \in \mathbb{R}^{L \times d}$ denote the query, key, and value obtained from the same input token embeddings in the $l$-th self-attention layer. The attention map is first computed as,

$$\mathbf{A}_l^{\text{self}} = \text{Softmax}\Big(\frac{\mathbf{Q}_l \mathbf{K}_l^\top}{\sqrt{d}}\Big), \tag{1}$$

which is then multiplied with the value to compute the output, $\mathbf{O}_l = \mathbf{A}_l^{\text{self}} \mathbf{V}_l$. Here, we would like to note that each of the output tokens $\mathbf{O}_l[i, :]$ is a *linear combination* of input token embeddings,

$$\underbrace{\mathbf{O}_l[i, :]}_{i^{\text{th}} \text{ output token}} = \sum_{j=1}^{L} \mathbf{A}_l^{\text{self}}[i, j] \underbrace{\mathbf{V}_l[j, :]}_{\text{input token}}, \tag{2}$$

where attention $\mathbf{A}_l^{\text{self}}[i, j]$ serves as the scalar coefficient of the $j$-th input token.

**Text-to-Image Personalization**. Text-to-image (T2I) personalization involves fine-tuning a pre-trained T2I model (Rombach et al., 2022) on a small set of personal concept images $\mathbf{x}$. An encoder $\mathcal{V}$ is adopted to produce latent image $\mathbf{z}_0 = \mathcal{V}(\mathbf{x})$, and a random noise map $\epsilon \sim \mathcal{N}(\mathbf{0}, \mathbf{I})$ is added to the latent $\mathbf{z}_t = \alpha_t \mathbf{z}_0 + \beta_t \epsilon$, where $\alpha_t$ and $\beta_t$ denote the noise scheduling coefficients (Ho et al., 2020). Also, a text prompt that includes a personal concept token representing the personal concept (*e.g., "a picture of a [v]"*) is tokenized as sequence of token embeddings $\mathbf{P} = \{p_1, \ldots, p_*, \ldots, p_L\}$, where $*$ denotes the index of the personal concept token. Utilizing token embeddings, the text embedding is obtained with CLIP (Radford et al., 2021) text encoder $\Gamma$, $\mathbf{C} = \Gamma(\mathbf{P})$. Leveraging the text embedding and the noisy latent, the T2I model $\epsilon_\theta$ is fine-tuned to minimize the following objective,

$$\mathcal{L}_{\text{Custom}}(\mathbf{z}_t, t, \mathbf{C}) := \mathbb{E}_{\mathbf{C}, \epsilon, t, \mathbf{z}}\Big[\|\epsilon - \epsilon_\theta(\mathbf{z}_t, t, \mathbf{C})\|_2^2\Big]. \tag{3}$$

### 2.2 LEARNING DIVERSE CONTEXTS IN TEXT SPACE

Existing T2I personalization (Gal et al., 2022; Ruiz et al., 2023; Kumari et al., 2023; Han et al., 2023) typically finetunes the pretrained T2I model on 4–5 images of the personal concept captured with limited contexts *e.g.,* simple background (Figure 1, (a)). This often leads to overfitting, leading to generated results with poor prompt fidelity (Figure 1, (b)). While this can be naively addressed by collecting images captured in diverse contexts, this is often impractical as images are costly to acquire (Figure 1, (c)).

**Context Diversification in Text Space.** Instead of diversifying the context within image space, we diversify the context within *text space* (Figure 1, (d)). Based on the fact that personal concepts are represented as tokens *e.g., "[v]"*, contextual diversification can be easily achieved within text space by simply combining the personal concept token with diverse contextual words, *e.g., "[v] at Amazon Forest"*. As manually collecting images of the personal concept in diverse contexts is costly, this provides a highly efficient way to improve the contextual diversity. Accordingly, we construct a prompt set that describes the personal concept token in diverse contexts prior to training (Figure 2, Diverse Contexts). The construction process is detailed in Section A.3.

**MLM for Learning Diverse Contexts.** Diversified contexts are learned within *text space* using masked language modeling (MLM) (Figure 1, (e)). T2I personalization typically requires images corresponding to each prompt for the denoising objective (Equation (3)), hence leveraging diversified text prompts for training is not straightforward. MLM can be an effective way to circumvent this restriction, since MLM operates entirely within *text space*. In MLM, the personal concepts are contextualized with diverse contexts via self-attention. By encouraging the model to predict the correct word, the personal concept is encouraged to be aligned with diverse semantics. The details of the learning process with MLM are elaborated below.

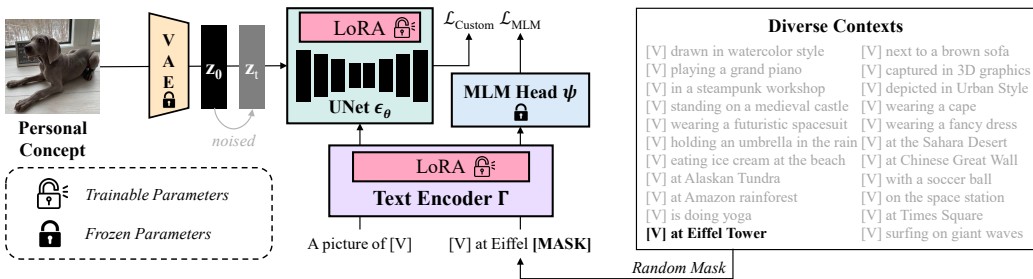

Figure 2: **Method Overview.** For T2I personalization, we fine-tune the model to align the personal concept image and the prompt containing the personal concept token *i.e., "a photo of [v]"*. During T2I personalization, we deploy a prompt set describing the personal concept, *i.e., "[v]"*, in diverse contexts. Sampling from the diversified prompt set, MLM loss is computed. In this way, diverse context learning can be achieved without requiring the paired image.

Tokenizing the prompt drawn from the contextually diverse prompt set (*e.g., "[v] at Eiffel Tower"*), we obtain a sequence of token embeddings $\widetilde{\mathbf{P}} = \{\widetilde{p}_i\}_{i=1}^{L}$, where $i$ denotes the index of each token and we denote the $\widetilde{p}_*$ as the personal concept token. For MLM, we randomly mask a subset of the tokens, and the mask token is denoted as $\widetilde{p}_m$. At each self-attention layer in the text encoder, the query, key, and value matrices are projected from the same input, denoted as $\mathbf{Q}_l$, $\mathbf{K}_l$, and $\mathbf{V}_l$, all in $\mathbb{R}^{L \times d}$. Attention map $\mathbf{A}_l^{\text{self}}$ is then computed (Equation (1)), which is then multiplied with the value $\mathbf{V}_l$. As discussed in Section 2.1, the output token is the *linear combination* of the input tokens,

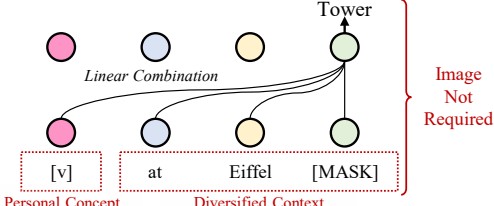

Figure 3: **MLM with Personal Concept.** The personal concept is contextualized with diverse contexts as a linear combination. The personal concept gets aligned with the contexts by learning to predict the correct masked token, notably without requiring an image.

$$\underbrace{\mathbf{O}_l[m,:]}_{\text{Output Mask}} = \sum_{j=1}^{L} \mathbf{A}_l^{\text{self}}[m,j]\mathbf{V}_l[j,:]$$

$$= \sum_{\substack{j=1 \\ j \neq *}}^{L} \mathbf{A}_l^{\text{self}}[m,j]\underbrace{\mathbf{V}_l[j,:]}_{\text{Context}} + \mathbf{A}_l^{\text{self}}[m,*]\underbrace{\mathbf{V}_l[*,:]}_{\text{Personal Concept}} ,$$

(4)

where $m$ denotes the index of mask token, $\mathbf{O}_l$ is the output of the $l$-th layer, and $\mathbf{V}_l[i,:] \in \mathbb{R}^d$ is the input token. As shown, when computing the output mask token, the personal concept is linearly combined with the context tokens. From contextualized mask token, masked tokens are predicted $\hat{\mathbf{y}} = \psi(\mathbf{O}_l[m,:])$. For correct prediction, the MLM loss is computed,

$$\mathcal{L}_{\text{MLM}} = \mathbb{E}[\text{CrossEntropy}(\mathbf{y}, \hat{\mathbf{y}})],$$

(5)

where $\psi$ is the MLM head for masked token prediction. As the CLIP (Radford et al., 2021) text encoder does not support MLM, we train a lightweight transformer network prior to the T2I personalization. The details are provided in Section A.4. Conceptually, minimizing the MLM loss encourages the personal concept embedding to learn the contextual semantics, as the output mask token incorporates the personal concept as a linear combination. The MLM process involving personal concept is illustrated in Figure 3. Note that neither the input nor the label requires *any image*.

**Finetuning for Personalization.** Our model personalization is based on DreamBooth (Ruiz et al., 2023) with LoRA (Hu et al., 2022) that achieves personalization with $\mathcal{L}_{\text{Custom}}$ (Equation (3)). Utilizing the contextually diverse text prompts, the $\mathcal{L}_{\text{MLM}}$ (Equation (5)) is computed. The overall learning objective can be formulated as follows,

$$\mathcal{L}_{\text{Custom}}(\mathbf{z}_t, t, \mathbf{C}) + \lambda \mathcal{L}_{\text{MLM}}(\widetilde{\mathbf{C}}_{\text{masked}}),$$

(6)

where $\mathbf{z}_t$ denotes the noised latent of personal concept image, $t$ denotes the timestep, and $\lambda$ denotes the weight for the MLM loss. Note that, as the MLM objective *does not* involve images, the model fine-tuning can be achieved with diverse contexts without manually collecting the corresponding images. We provide the illustrative description of the overall training process in Figure 2.

## 2.3 ANALYSIS

We provide an analysis of our method to explain how the MLM with diverse contexts during model personalization achieves improvements. We empirically validate that personalization with diverse contexts leads to enhancement in *textual space*, and this further leads to enhancement in *image space* with improved prompt fidelity in T2I generation.

**Analysis in Textual Space.** Here, we show that the T2I personalization trained with *limited contexts* leads to suppression of context tokens and dominance of personal concepts. In contrast, our method customized with *diverse contexts* mitigates this issue, thereby achieving semantic enhancement in textual space. To show this, we validate the following,

- The semantics of *context tokens* (*i.e.*, non-personal) become *similar* to the *personal concept tokens* when customized with limited contexts.

- The semantics of the *context tokens* get *distinct* from the *personal concept* token as the model is customized with diverse contexts.

To validate this, we provide a cosine distance analysis of text token embeddings. Specifically, we measure the cosine distance between the context token $c_b$, *e.g.,* *"Eiffel Tower"* and the personal concept token embedding $c_*$, *e.g.,* *"[v]"*. The token embeddings are extracted from the output embedding of the same prompt, *e.g.,* *"a photo of [v] at Eiffel Tower"*, and their cosine distance $d_{\text{text}} = 1 - \cos(c_b, c_*)$ is measured to see how distinct they are. For validation, we use 100 different context tokens with 7 different concepts.

| Method | $d_{\text{text}} \uparrow$ | $SV_{\text{cross}} \downarrow$ |
|---|---|---|
| w/o MLM (Limited Contexts) | 0.39 | 23.27 |
| Ours (Diverse Contexts) | **0.74** | **15.18** |

Table 1: **Analysis.** Cosine distance between the personal concept and context token from the same prompt ($d_{\text{text}}$) is measured. $SV_{\text{cross}}$ denotes spatial variance of the cross-attention maps of the context token.

The result in Table 1 shows that the context token embeddings in the model trained without MLM with limited contexts become highly similar to the personal concept (*low* $d_{\text{text}}$), whereas we observe the opposite trend for our model trained with diverse contexts (*high* $d_{\text{text}}$). This result suggests that limited-context personalization induces a collapse of contextual semantics, while diverse contexts effectively mitigate this, keeping the context tokens semantically distinct.

**Analysis in Image Space.** In order to see how the enhancement in textual space can be transferred to image space, we provide an analysis of the cross-attention layers in the diffusion U-Net, where the interaction between text and image takes place.

Let $\mathbf{Q}_{\mathcal{I}} \in \mathbb{R}^{|\text{queries}| \times d}$ and $\mathbf{K}_{\mathcal{T}} \in \mathbb{R}^{L \times d}$ denote the query and key in the cross-attention layer, projected from image $\mathcal{I}$, and text $\mathcal{T}$, where $|\text{queries}|$ denotes the number of image tokens, $L$ denotes the number of text tokens, and $d$ denotes the dimension of each image/text token embedding. In each layer, the cross-attention

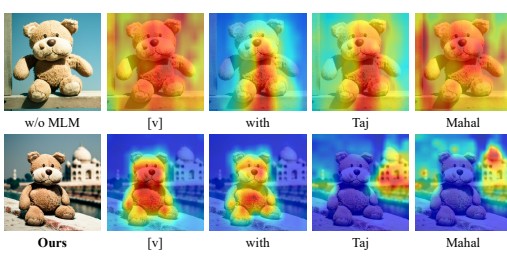

Figure 4: **Cross-Attention Map Visualization.** With our approach, attention maps of context tokens are more concentrated within relevant regions. More results available in Figures G and H. $16 \times 16$ map is visualized.

map is computed as,

$$\mathbf{A}_l^{\text{cross}} = \text{Softmax}\left(\frac{\mathbf{Q}_{\mathcal{I}}\mathbf{K}_{\mathcal{T}}^{\top}}{\sqrt{d}}\right). \tag{7}$$

We visualize the cross-attention map obtained from the model trained without MLM and our model trained with MLM in Figure 4. For the model trained without MLM, the visualization shows that the similarities in the text embeddings are transferred to the image space, resulting in cross-attention maps of the context tokens being dispersed over the image. In contrast, our method with MLM shows attention maps being more concentrated within the relevant regions that are distinct from the personal concept. To assess the overall trend of this observation, we analyze cross-attention maps using the same prompt set as the textual analysis. For this, we compute the Spatial Variance Gonzalez (2009) ($\text{SV}_{\text{cross}}$) of cross-attention map to measure the dispersion of activations around their centroid,

$$\mu_j = \sum_{i,j} j\mathbf{A}_l^{\text{cross}}[i,j], \quad \mu_i = \sum_{i,j} i\mathbf{A}_l^{\text{cross}}[i,j]$$

$$\text{Var}_j = \sum_{i,j} \mathbf{A}_l^{\text{cross}}[i,j](j-\mu_j)^2, \quad \text{Var}_i = \sum_{i,j} \mathbf{A}_l^{\text{cross}}[i,j](i-\mu_i)^2, \tag{8}$$

$$\text{SV}_{\text{cross}} = \tfrac{1}{2}\left(\text{Var}_j + \text{Var}_i\right).$$

As shown in Table 1, the model trained without MLM yields higher spatial variance, indicating that cross-attention is dispersed over the image, aligned with the visualized example in Figure 4. In contrast, our model trained with MLM yields lower spatial variance, indicating that cross-attention is more tightly confined to relevant regions, aligned with the visualized example in Figure 4, (*"Taj Mahal"*). This result indicates that semantic enhancement in textual space leads to enhancement in image space, confirming the effectiveness of the proposed method. Additional cross-attention maps are visualized in Figures G and H.

## 3 EXPERIMENTS

### 3.1 EXPERIMENTAL SETTINGS

**Implementation Details.** We fine-tune the pretrained Stable Diffusion (Rombach et al., 2022) `stabilityai/stable-diffusion-2-1-base` for personalization. For MLM, we randomly mask tokens with 15% chance, following (Devlin et al., 2018). For fine-tuning with MLM loss, we *strictly* exclude the benchmark prompts from the training data. We set classifier-free guidance (Ho & Salimans, 2022) scale of 7.5 with the number of sampling steps 50 for generation. We set LoRA rank=32, and the AdamW (Loshchilov, 2019) is adopted as optimizer. All the experiments are conducted on a single NVIDIA RTX 3090 24GB GPU.

**Baselines.** We compare our method with Textual Inversion (TI) (Gal et al., 2022), DreamBooth (DB) (Ruiz et al., 2023), XTI (Voynov et al., 2023), NeTI (Alaluf et al., 2023), ELITE (Wei et al., 2023), BLIP-D (Li et al., 2023a), CustomDiffusion (CD) (Kumari et al., 2023), and DisenBooth (Chen et al., 2023). We provide details of the baselines in Section A.1.

**Dataset and Benchmark.** We use a mixture of 15 different personal concepts adopted from DB (Ruiz et al., 2023), TI (Gal et al., 2022), and CD (Kumari et al., 2023). Specifically, we use 11 subjects from DB (Ruiz et al., 2023) composed, 3 subjects from (Kumari et al., 2023), wooden_pot], and 1 subject from (Gal et al., 2022). We use a benchmark prompt set from DB (Ruiz et al., 2023). This prompt set contains 25 prompts for nonliving and living subject categories, and 8 images per prompt are generated, resulting in a total of 3,000 images. Also, we use a more challenging prompt set from (Nam et al., 2024). This prompt set consists of 24 prompts for both nonliving and living subject categories. We generate 8 images per prompt, resulting in a total of 2,880 images.

**Evaluation Metrics.** Following the literature (Ruiz et al., 2023; Gal et al., 2022; Kumari et al., 2023; Kim et al., 2024b), we measure: 1) CLIP Text Alignment (Hessel et al., 2021) (**CLIP**) to measure the text prompt fidelity, 2) DINO Image alignment (**DINO**) to measure the subject fidelity. We mask the non-subject regions when computing this, following (Kim et al., 2024b). Additionally, we evaluate 3) **ImageReward**, which assesses the overall aesthetic and alignment quality of generated images

| Method | CLIP ↑ | DINO ↑ | ImageReward ↑ | BLIP-VQA ↑ |
|---|---|---|---|---|
| TI | 0.682 | 0.708 | -0.769 | 0.372 |
| NeTI | 0.733 | 0.754 | 0.010 | 0.545 |
| ELITE | 0.734 | 0.693 | 0.069 | 0.594 |
| BLIP-D | 0.745 | 0.687 | 0.206 | 0.607 |
| CD | 0.766 | 0.750 | 0.569 | 0.678 |
| DB | 0.728 | 0.744 | 0.036 | 0.544 |
| XTI | 0.748 | 0.721 | 0.414 | 0.692 |
| DisenBooth | 0.771 | 0.743 | 0.636 | 0.640 |
| Ours | **0.785** | **0.760** | **0.842** | **0.726** |

Table 2: **Comparison results on DreamBench prompts.** The best results are denoted in **bold**.

| Method | CLIP ↑ | DINO ↑ | ImageReward ↑ | BLIP-VQA ↑ |
|---|---|---|---|---|
| TI | 0.689 | 0.705 | -0.688 | 0.377 |
| NeTI | 0.735 | 0.746 | -0.092 | 0.524 |
| ELITE | 0.746 | 0.683 | 0.064 | 0.581 |
| BLIP-D | 0.741 | 0.692 | 0.084 | 0.571 |
| CD | 0.763 | 0.743 | 0.414 | 0.610 |
| DB | 0.747 | 0.740 | 0.146 | 0.556 |
| XTI | 0.756 | 0.703 | 0.269 | 0.591 |
| DisenBooth | 0.767 | 0.749 | 0.416 | 0.587 |
| Ours | **0.777** | **0.759** | **0.572** | **0.635** |

Table 3: **Comparison results on challenging prompts.** The best results are denoted in **bold**.

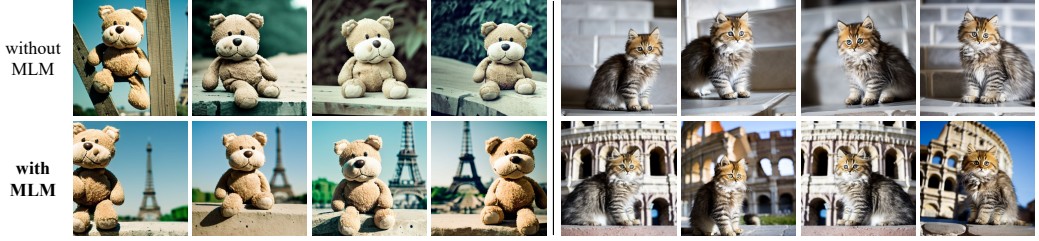

With Eiffel Tower in the background          With Roman Colosseum in the background

Figure 5: **Qualitative Comparison in the Ablation Study.** Training the model with diverse contexts with MLM leads to notable improvements in text-image alignment, while preserving the subject identity.

based on human preferences (Xu et al., 2023), and 4) **BLIP-VQA**, a widely used vision-language question-answering metric that verifies the presence of expected attributes in the generated images by answering predefined queries (Huang et al., 2023).

## 3.2 QUANTITATIVE RESULTS

**Comparison with Baseline Method.** Across both DreamBench (Table 2) and the challenging prompt set (Table 3), our method consistently achieves the best performance on all four metrics, demonstrating strong robustness and effectiveness. Improvements across both benchmarks further confirm its reliability in handling diverse prompt complexities. Notably, the substantial gain in ImageReward indicates that our approach not only preserves semantic alignment with the prompts but also generates outputs that better match human visual preferences. We also would like to note that this result is completely zero-shot, as we do not involve *any* of the benchmark prompts for the MLM.

## 3.3 ABLATION STUDY

In this section, we conduct ablation studies to provide deeper insights into our method and validate its effectiveness. For this, we use a subset of eight different concepts.

**Impact of Proposed Approach.** We first examine the impact of the proposed method. As presented in Table 4, incorporating diverse contexts during training enhances semantic alignment, resulting in substantial gains in ImageReward and BLIP-VQA. We also provide a qualitative comparison to visualize the impact of the MLM in Figure 5. As shown in the results, our approach trained with diverse contexts leads to higher text-image alignment Section B.1.

| $\lambda$ | ImageReward ↑ | BLIP-VQA ↑ | DINO ↑ |
|---|---|---|---|
| w/o MLM | 0.562 | 0.651 | 0.791 |
| $1 \times 10^{-2}$ | 0.728 | 0.680 | 0.794 |
| $\mathbf{2 \times 10^{-2}}$ | **0.781** | **0.694** | **0.790** |
| $3 \times 10^{-2}$ | 0.751 | 0.685 | 0.791 |
| $5 \times 10^{-2}$ | 0.862 | 0.700 | 0.776 |

Table 4: **Evaluation results with Varying $\lambda$.** Ours choice is **bolded**.

| Mask Prob. [%] | ImageReward ↑ |
|---|---|
| w/o MLM | 0.562 |
| 1 | 0.708 |
| 15 | **0.759** |
| 25 | 0.754 |
| 45 | 0.750 |

| # Prompts | ImageReward ↑ | BLIP-VQA ↑ |
|---|---|---|
| w/o MLM | 0.562 | 0.651 |
| 1 | 0.418 | 0.603 |
| 100 | 0.751 | 0.678 |
| 300 | 0.756 | 0.684 |
| Ours | **0.781** | **0.694** |

Table 5: **Ablation Study on Masking Probability.** Table 6: **Ablation Study on Prompt Set Size.**

**Impact of $\lambda$.** We next study the influence of MLM loss weight (Table 4). By adopting the MLM, we observe notable improvements in the ImageReward score. By increasing the value of $\lambda$, we observe a general trend of improvement in ImageReward. Besides, an excessive amount of value results in a decrease in subject fidelity, as the model is encouraged to focus more on capturing the semantic relationships of the contexts. We use $\lambda = 2 \times 10^{-2}$ as final choice.

**Impact of Masking Probability.** We train the model with different masking probabilities and study its impact. Table 5 shows that the performance is relatively insensitive to the masking probability; however, excessively high values lead to degradation. This result validates the importance of contextual information, as excessively high masking value leads to contextual semantics removal.

**Impact of Prompt Set Size.** We study the impact of prompt set size when applying the MLM objective, by varying their sizes (Table 6). Here, we sample different numbers of prompts from each text prompt category. We observe that a very small prompt set (e.g., size = 1) fails to provide effective contextualization, whereas increasing the prompt size improves the model's ability to follow the prompts more effectively.

## 3.4 QUALITATIVE RESULTS

We present the side-by-side comparison results of the proposed method in Figure 6. In general, the baseline method that integrates our approach leads to improvement in prompt fidelity. In these visualization results, the baseline approach shows a higher tendency to neglect the context semantics. We analyze that the baseline approach loses the semantics of the context in textual space, which leads to the loss of semantics of the generated images. In contrast, the adoption of MLM leads to the preservation of the contextual semantics in text embedding, resulting in images with enhanced semantics with higher prompt fidelity. We provide additional qualitative results in Section B.

## 4 RELATED WORKS

**Text-to-Image Generation.** The emergence of diffusion models (Ho et al., 2020; Nichol & Dhariwal, 2021; Song et al., 2020a;b; 2023; Dhariwal & Nichol, 2021) have shifted the paradigm of generative modeling, with its capabilities demonstrated in a wide range of computer vision applications, including videos (Ho et al., 2022; Jain et al., 2024; Kim et al., 2023; Wei et al., 2024), segmentation (Tian et al., 2024; Peng et al., 2023a;b), and super resolutions (Yang et al., 2024; Xia et al., 2023; Feng et al., 2024). Among them, diffusion-based text-to-image (T2I) generative models (Saharia et al., 2022; Rombach et al., 2022; Balaji et al., 2022; Ramesh et al., 2021; 2022; Ding et al., 2022) have achieved significant success. GLIDE (Nichol et al., 2021) demonstrated that using classifier-free guidance (Ho & Salimans, 2022) can enhance both the photorealism and caption alignment of generated images. Stable Diffusion (Rombach et al., 2022) and its variants (Podell et al., 2023; Esser et al., 2024) proposed a way to improve the efficiency by applying the diffusion process in the lower-dimensional latent space. Several research efforts have been introduced to improve the controllability of T2I generation across various dimensions by incorporating additional input as conditions (Zhang et al., 2023; Li et al., 2024b; Dahary et al., 2024; Li et al., 2023b; Gafni et al., 2022; Tuo et al., 2023; Chen et al., 2024a).

**Personalized Text-to-Image Genearation.** Building on the great success of T2I generation, a line of research works has been proposed to adapt the T2I generative model on user-provided custom images (Gal et al., 2022; Ruiz et al., 2023; Kumari et al., 2023; Voynov et al., 2023; Han et al., 2023; Liu et al., 2023; Nam et al., 2024; Po et al., 2024; Alaluf et al., 2023). Textual inversion (TI) (Gal et al., 2022) has pioneered a method to convert personal concept images into token embedding. DreamBooth (Ruiz et al., 2023) extended TI by fine-tuning the diffusion U-Net along with the prior-preservation loss to prevent the forgetting of prior concepts. Since the introduction of the pioneering

| Concept | CD | DB | DisenBooth | XTI | BLIP-D | **Ours** |
|---|---|---|---|---|---|---|

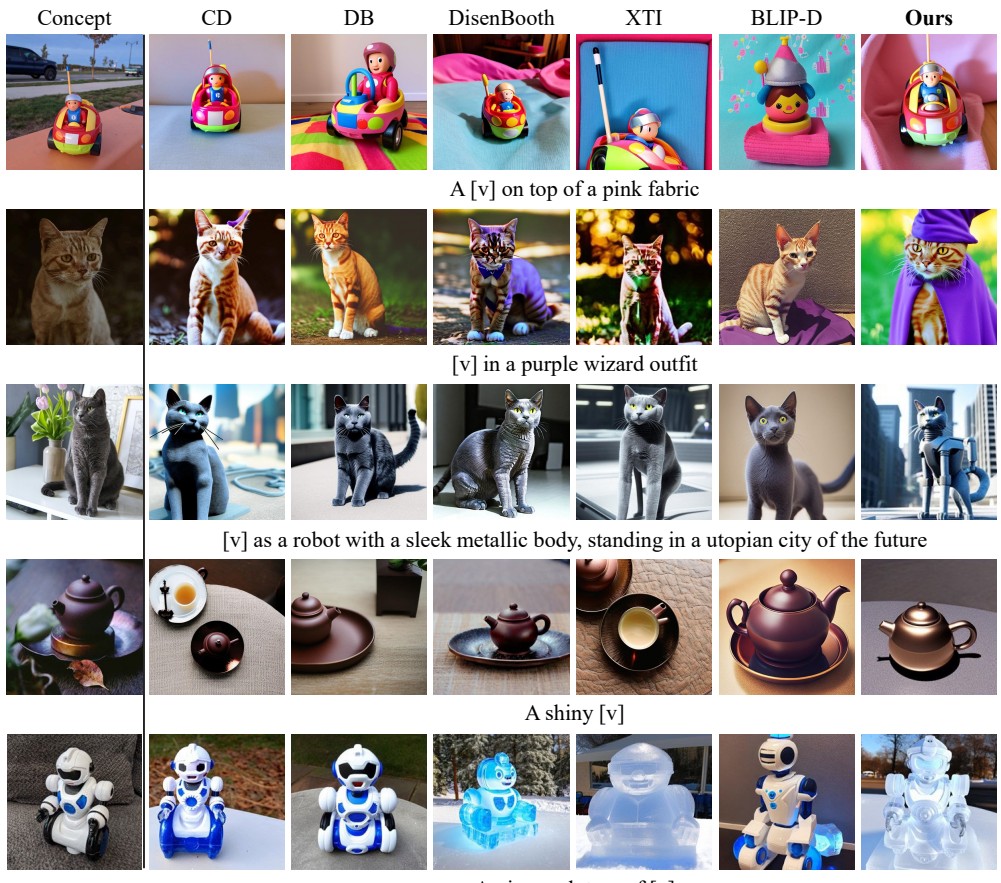

A [v] on top of a pink fabric

[v] in a purple wizard outfit

[v] as a robot with a sleek metallic body, standing in a utopian city of the future

A shiny [v]

An ice sculpture of [v]

Figure 6: **Qualitative Comparisons Results.** Our method shows clear superiority in text–image semantic alignment and preserves subject identity. In contrast, existing methods often yield low text-image alignment or compromise identity when semantically aligned (e.g., *DisenBooth* and *XTI* in the last row).

works, a line of work has been proposed to make improvements in diverse aspects. Many works (Shi et al., 2024; Gal et al., 2023; Arar et al., 2023; Jia et al., 2023; Li et al., 2024a; Zhang et al., 2024) such as InstantBooth have focused on the fast adaptation of the model without test-time optimization. There is also a growing interest in developing methods specialized for facial images (Peng et al., 2024; Pang et al., 2024; Kim et al., 2024a; Wang et al., 2024; Gu et al., 2024), such as (Yuan et al., 2023) which constructed a set of basis tokens corresponding to celebrities and optimized their weights to synthesize a given image. Our work also has close relations to works that aim to improve the semantic alignment in T2I generation (Kim et al., 2024b; Chen et al., 2023). For example, (Kim et al., 2024b) proposed to make improvements by providing selectively informative captions during training.

## 5 CONCLUSION

In this paper, we proposed a novel text-to-image (T2I) personalization approach for robust finetuning of the T2I diffusion model. Unlike prior methods that rely on collecting additional images to address the limited contextual diversity, our approach offered an efficient way to achieve this, enhancing the diversity of a personal concept's context within textual space. We show that this increased diversity in text space can be effectively learned with masked language modeling, demonstrated by extensive experiments with state-of-the-art results on public benchmarks. With further analysis showing that context diversification enhances textual representations and propagates to image space via cross-attention mechanisms, we provided insight into why our approach leads to improved personalization in T2I generation.

## ETHICS STATEMENT

Our research investigates customized text-to-image (T2I) generative models. We recognize that such models entail potential risks, including misuse for creating misleading, harmful, or biased content. These risks may manifest in the generation of deceptive imagery, reinforcement of stereotypes, or the production of offensive or inappropriate material in certain contexts. While the primary aim of this work is academic, we acknowledge the broader ethical implications and societal impact associated with advancing and deploying generative AI technologies. We underscore the need for safeguards that promote the safe and ethical use of these models.

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

# A  ADDITIONAL DETAILS

## A.1  ADDITIONAL DETAILS OF BASELINES

**Textual Inversion (Gal et al., 2022).**   We train the model for 5,000 iterations with learning rate `2e-2`. We use a batch size of 8 to train the method. Other than the personal concept token embeddings, no parameters are updated during the training.

**XTI (Voynov et al., 2023).**   We follow the configurations from (Gal et al., 2022). We train the model for 500 iterations with a constant learning rate of `2e-3`. We use a batch size of 4 to train the method. Following the original paper, different special token embeddings are learned for each layer in the UNet.

**DreamBooth (Ruiz et al., 2023).**   For this baseline method, we train the model for 1,000 iterations with a constant learning rate of `1e-6`. We use a batch size of 1 to train the method. Following the original method, the prior preservation loss is adopted during the training. For this, we generate a set of 200 images by prompting SD (Rombach et al., 2022) with *"a picture of [SUBJECT CLASS]"*, where *'[SUBJECT]'* is replaced by the name of the class corresponding to the personal concept. During customization, we update all the parameters including the CLIP text encoder and the diffusion U-Net.

**CustomDiffusion (Kumari et al., 2023).**   For this baseline method, we train the model for 500 iterations with a constant learning rate of `4e-5`. We use a batch size of 4 to train the method. Following the original method, we adopt prior preservation loss during training. For this, we use the same 200 generated images used for DB. During training, only the Key/Value projection layers of the diffusion U-Net are updated.

**DisenBooth (Chen et al., 2023).**   We fine-tune the model for 3,000 iterations with a learning rate of `1e-4` and a batch size of 1, following the paper. In this method, LoRA (Hu et al., 2022) is adopted for fine-tuning the UNet.

**NeTI (Alaluf et al., 2023).**   To customize with NeTI (Alaluf et al., 2023), we fine-tune the model for 250 iterations with a learning rate of `1e-3` and a batch size of 8, adhering to the original configuration. The parameters of diffusion UNet remain fixed during fine-tuning.

**ELITE (Wei et al., 2023) & BLIP-D (Li et al., 2023a).**   We utilize the official GitHub implementations provided by the authors.

---

**Algorithm 1** Customization Process with MLM

---

1: Load parameters $\theta$, $\psi$, $\Gamma$, and $\mathcal{V}$         $\{\theta$: U-Net, $\psi$: MLM Head, $\Gamma$: text encoder, $\mathcal{V}$: VAE$\}$
2: Fix $\psi$ and $p_m$         $\{p_m$: mask embedding$\}$
3: Set $\lambda$         $\{\lambda$: MLM weight$\}$
4: Initialize LoRA params $\Theta$
5: **repeat**
6:     Sample $t \sim \text{Uniform}[0, \text{T} - 1], \epsilon \sim \mathcal{N}(\mathbf{0}, \mathbf{I})$
7:     Encode $\mathbf{z}_0 = \mathcal{V}(\mathbf{x})$, $\mathbf{C} = \Gamma(\mathbf{P})$ and $\widetilde{\mathbf{C}} = \Gamma(\widetilde{\mathbf{P}})$
8:     Get noised latent, $\mathbf{z}_t = \alpha_t \mathbf{z}_0 + \beta_t \epsilon$
9:     Compute $\mathcal{L}_{\text{Diff}} = \mathbb{E}_{\mathbf{C}, \epsilon, t, \mathbf{z}} || \epsilon - \epsilon_\theta(\mathbf{z}_t, t, \mathbf{C}) ||_2^2$
10:    Compute $\mathcal{L}_{\text{MLM}} = \mathbb{E}_{\mathbf{y}, \mathbf{P}_{\text{masked}}} \left[ \text{CrossEntropy}(\mathbf{y}, \Gamma(\mathbf{P}_{\text{masked}})) \right]$
11:    Gradient descent optimization on $\nabla_\Theta \left[ \mathcal{L}_{\text{Diff}} + \lambda \mathcal{L}_{\text{MLM}} \right]$
12: **until** optimized

---

## A.2  DETAILS OF EVALUATION METRICS

**DINO Score.**   To evaluate subject alignment, we calculate the DINO image similarity between the generated image and the reference image. Following (Kim et al., 2024b), we measure the DINO

similarity using the masked foreground region of the subject. This metric is denoted as $I_{\text{DINO-FG}}$. DINO embeddings are obtained from ViT-S/16. Binary segmentation masks are generated using Grounded-SAM, conditioned on the subject's prior concept name *e.g., "a dog"*. Examples of binary masks and the corresponding masked images are shown in Figure A.

**CLIP Score.** To evaluate text alignment, we calculate the CLIP image-text similarity between the generated image and the input prompt, denoted as $T_{\text{CLIP}}$. When encoding the text prompt, the personal concept token is removed from the prompt and only the prior concept token is retained, *e.g., a picture of a cat in the snow*. We obtain the CLIP embeddings from ViT-B/32. This measure does not involve the use of segmentation masks.

**ImageReward.** We employ ImageReward (Xu et al., 2023) to evaluate the aesthetic and semantic quality of generated images. We utilize a reward model pretrained to align human aesthetic preferences, considering factors such as composition, coherence, and prompt fidelity. The reward model processes both the generated image and the corresponding text prompt to output a scalar reward score, which we use as the Image Reward Score. The scores are obtained using the publicly available ImageReward model.

**BLIP-VQA.** This metric assesses the attribute binding in generated images (Huang et al., 2023). Specifically, the input prompt is broken down into multiple questions, which are then fed into a pretrained BLIP-VQA model along with the generated images. The model's probability of responding "yes" serves as the score for each question.

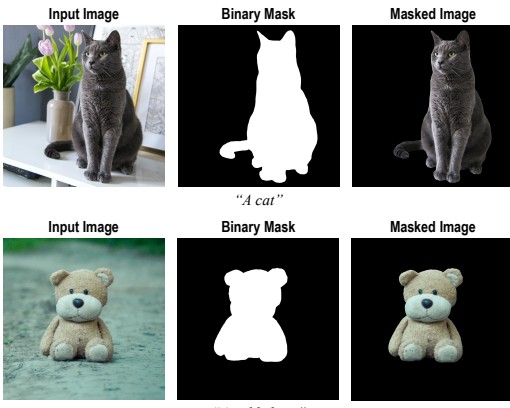

Figure A: **Example segmentation results obtained with Grounded-SAM.** We present triplets consisting of the input image, the binary mask, and the resulting masked image. The conditioning prompt for Grounded-SAM is indicated below each triplet. The masked images are utilized to compute the subject alignment score, $I_{\text{DINO-FG}}$.

### A.3 DETAILS OF TEXT PROMPT SET CONSTRUCTION

We describe the details of the prompt set construction process. The constructed prompts are used for masked language modeling (MLM). The prompt set comprises two types: living object prompts and non-living object prompts. For each prompt type, we predefined five different types of context. For each context type, we define different prompt templates and use different contextual words to diversify the prompt. To construct the corresponding contextual word set, we query the large language model (OpenAI, 2023) and insert it into the template.

**Living Object Prompts.** The predefined context types for living objects are shown below. In total, we construct a prompt set of size 38,394. During model customization, prompts are evenly sampled from each context type to construct a balanced batch.

1. **Human Interactive Prompts:** This prompt set includes various interactions involving human subjects. The example template that we use is *"[HUMAN_SUBJECT] is [INTER-ACTION] [CONCEPT]"*. For instance, *"Albert Einstein is watching TV with [v] dog"*. We use 49 human interaction terms and 72 human subject terms to be inserted into the template. In total, we generate 3,528 text prompts.

2. **Relative Position Prompts:** This prompt set includes scenarios where the personal concept token is combined with various positioning terms, colors, and objects. The example template that we use is *"[CONCEPT] [RELATIVE] a [COLOR] [OBJECT]"*. For instance, *"[v] dog next to a red vase"*. We use 18 relative terms, 24 color terms, and 68 object terms to be inserted into the template. In total, we generate 29,376 text prompts.

3. **Background Prompts:** This prompt set includes scenarios where the personal concept token is combined with various background terms. The example template that we use is *"[CONCEPT] with the [BACKGROUND] in the background"*. For instance, *"[v] dog with Eiffel Tower in the background"*. We use 138 background terms. In total, we generate 552 text prompts.

4. **Image Style Prompts:** This set of prompts includes scenarios where the personal concept token is combined with various terms that describe the overall style of the image. The example template that we use is *"[CONCEPT] depicted in [STYLE] style"*. For instance, *"[v] dog in Pop Art style"*. We use 126 style terms with 3 different templates. In total, we generate 378 prompts.

5. **Attribute Changing Prompts:** This set of prompts includes scenarios where the personal concept token is combined with various outfit terms. The example template that we use is *"[CONCEPT] in a [COLOR] [OUTFIT]"*. For instance, *"[v] dog in a red cowboy outfit"*. We use 162 outfit terms with 24 color terms. In total, we generate 4,560 prompts.

**Non-Living Object Prompts.** The predefined context types for non-living objects are shown below. In total, we construct a prompt set of size 34,599. During model customization, prompts are evenly sampled from each context type to construct a balanced batch.

1. **Human Interactive Prompts:** This prompt set includes various interactions involving human subjects. The example template that we use is *"[HUMAN_SUBJECT] is [INTERAC-TION] [CONCEPT]"*. For instance, *"Einstein is holding a [v] pot"*. We use 58 human interaction terms and 72 human subject terms to be inserted into the template. In total, we generate 4,176 prompts.

2. **Relative Position Prompts:** This prompt set includes scenarios where the personal concept token is combined with various positioning terms, colors, and objects. The example template that we use is *"[CONCEPT] [RELATIVE] a [COLOR] [OBJECT]"*. For instance, *"[v] pot on a beige sofa"*. We use 18 relative terms, 24 color terms, and 68 object terms to be inserted into the template. In total, we generate 29,376 prompts.

3. **Background Prompts:** This prompt set includes scenarios where the personal concept token is combined with various background terms. The example template that we use is *"[CONCEPT] with the [BACKGROUND] in the background"*. For instance, *"[v] pot with forest in the background"*. We use 138 background terms. In total, we generate 552 prompts.

4. **Image Style Prompts:** This set of prompts includes scenarios where the personal concept token is combined with various terms that describe the overall style of the image. The example template that we use is *"[CONCEPT] depicted in [STYLE] style"*. For instance, *"[v] dog in 3D rendering style"*. We use 126 style terms with 3 different templates. In total, we generate 378 prompts.

5. **Attribute Changing Prompts:** This set of prompts includes scenarios where the personal concept token is combined with various attributes to be bonded, such as colors and shapes. The example template that we use is *"a [CONCEPT] in a [ATTRIBUTE] shape"*. For instance, *"a [v] pot in a triangular shape"*. We use 75 attribute terms. In total, we generate 117 prompts.

## A.4 DETAILS OF MLM HEAD

As the CLIP (Radford et al., 2021) text encoder does not support MLM, we train a transformer network (MLM Network) $\psi$ prior to the T2I personalization. During the pretraining, the personal concept token is not involved. We only train the mask embedding and the self-attention layers in $\psi$. The others, including the CLIP text encoder non-mask tokens, remain fixed.

**Architecture.** The MLM Head consists of four blocks of a self-attention layer and a feed-forward layer, followed by a layer normalization layer, where each block learns the residuals of the input with the residual connection. We set the hidden dimension as 768 for all layers.

**Training.** To train the MLM Head, we use a merged set of COCO caption datasets and the prompt set that we constructed. For the manual prompt set, we replace the personal concept token with the personal concept token to the corresponding prior concept token. During training, we set the ratio of the batch of the two prompt sets to be 70 to 30. No personal concept token embeddings are involved during the contextualization pretraining. The MLM Head is pretrained for 100K iterations with a learning rate of $1e-4$, and batch size 150. We use the AdamW optimizer for training. The detailed pretraining procedure is described in Algorithm 2.

---

**Algorithm 2** Training Procedure of MLM Head

---

1: Load parameters $\Gamma$                                                                                      {$\Gamma$: CLIP text encoder}
2: Initialize $\psi$, $p_{\text{mask}}$                                                        {$\psi$: MLM Head, $p_{\text{mask}}$: mask embedding}
3: Set $\rho_{\text{mask}}$                                                                      {$\rho_{\text{mask}}$: masking probability}
4: **repeat**
5:     Sample $\mathbf{t}$ from rich prompt set, $\mathbf{P} = \text{Tokenize}(\mathbf{t})$
6:     $\mathbf{P}_{\text{masked}}, \mathbf{y} = \text{RandomMask}(\mathbf{P}, \rho_{\text{mask}})$                            {$\mathbf{y}$: masked token label}
7:     Compute $\mathcal{L}_{\text{MLM}} = \mathbb{E}_{\mathbf{y}, \mathbf{P}_{\text{masked}}}[\text{CrossEntropy}(\mathbf{y}, \psi(\Gamma(\mathbf{P}_{\text{masked}})))]$
8:     Optimize $\nabla_{\psi, p_{\text{mask}}} \mathcal{L}_{\text{MLM}}$ with gradient descent
9: **until** convergence

---

## A.5 DETAILS OF ANALYTICAL STUDY

We provide additional details of the analytical experiment we described in Section 2.3.

**Prompt Set Construction for Analytical Study.** The prompt set for our analysis consists of 100 text prompts for each of the 7 concepts, totaling 700 prompts. We use two templates: *a picture of [SUBJECT] with the [CONTEXT] in the background"* and *a picture of [SUBJECT] with the [PREPOSITION] [OBJECT]"*. Here, *'[SUBJECT]'* is replaced by the personal concept token; *[PREPOSITION]* is a spatial relation (e.g., *next to*); and *[OBJECT]* is an object name. We obtain a list of 50 background terms by querying a large language model (OpenAI, 2023) and substitute *[CONTEXT]* with these terms. The remaining 50 prompts per concept are generated directly by the language model. The complete list of prompts appears in Figure F. This prompt set is used for all experiments reported in Table 1 of Section 2.3.

## A.6 DETAILS OF DATASETS

**Personal Concepts.** We use a mixture of 15 different subjects adopted from DB (Ruiz et al., 2023), TI (Gal et al., 2022) and CD (Kumari et al., 2023). Specifically, we use 11 subjects from DB (Ruiz et al., 2023) composed of: [backpack, backpack_dog, cat, cat2, dog3, dog6, duck_toy, poop_emoji, rc_car, teapot, pushie_teddybear], and we use 3 subjects from CD (Kumari et al., 2023): [pet_cat1, pet_dog1, decoritems_woodenpot]. Finally, we use 1 subject from TI (Gal et al., 2022): [cat toy].

**Benchmark Prompts.** For generations, we utilize a benchmark prompt set from DreamBench (Ruiz et al., 2023), and (Nam et al., 2024). DreamBench contains 25 prompts for nonliving and living subject categories. The prompt set for non-living objects comprises 20 contextualization prompts and 5 property modification prompts. The prompt set for living objects contains 10 contextualization prompts, 10 accessorization prompts, and 5 property modification prompts. We generate 8 images

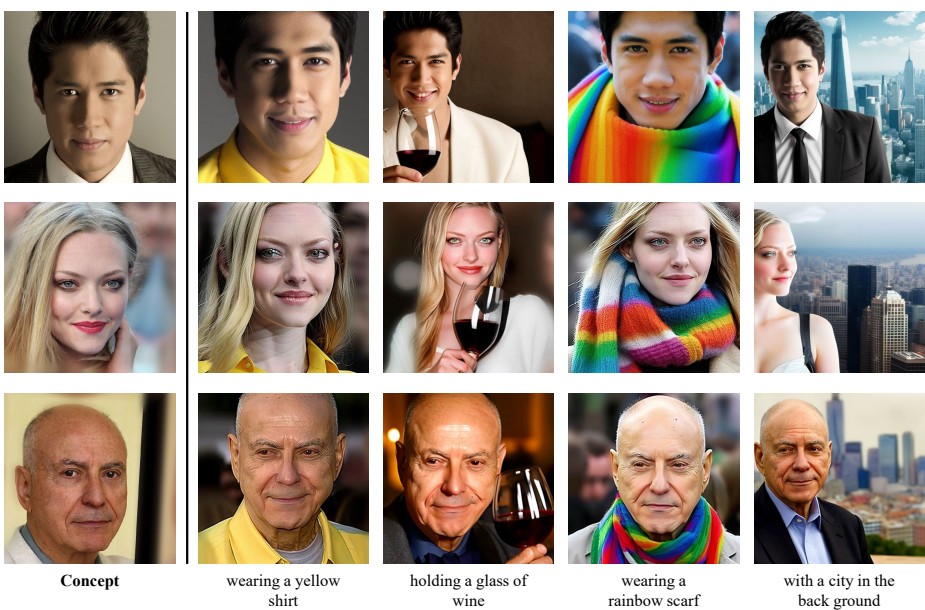

| Concept | wearing a yellow shirt | holding a glass of wine | wearing a rainbow scarf | with a city in the back ground |

Figure B: **Qualitative Results on Face Images.** The proposed method effective personalizes of fine-grained face-concept images.

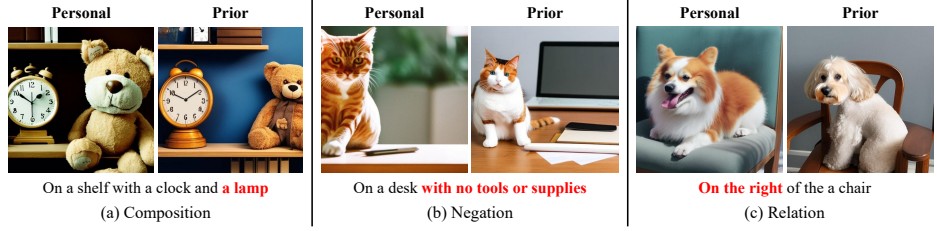

| On a shelf with a clock and **a lamp** | On a desk **with no tools or supplies** | **On the right** of the a chair |
| (a) Composition | (b) Negation | (c) Relation |

Figure C: **Failure Cases.** Failure cases obtained from (a) composition, (b) negation, and (c) relation. Each prompt is shown with output from a non-personalized model (prior), illustrating that the failures in the personalized results stem from the base model.

per prompt, yielding a total of 3,000 images. (Nam et al., 2024) provides more challenging prompts, containing 24 prompts for both living and non-living objects. The prompt set consists of 10 prompts for large displacements, 10 prompts for occlusion, and 4 prompts for novel-view synthesis.

# B ADDITIONAL RESULTS

## B.1 ADDITIONAL QUALITATIVE EXAMPLES

**Face Images.** We present personalization results for face images in Figure B. The results indicate that the proposed method is effective not only for coarse-grained subjects but also for capturing fine-grained individual traits while preserving overall visual quality. The personalized outputs appear natural and remain coherent with the input prompt.

**Failure Cases.** We report failure cases of our personalized concept model when evaluated on compositional, relational, and negation prompts. In our analysis, objects included in compositional prompts often fail to appear *e.g.,* no lamp in Figure C, or in the negation prompt, the negative objects still appear in the generation, *e.g.,* the negative object, tools still appear in the image or in the relation prompt, the subjects are generated in an incorrect relation to other objects, *e.g.,* a dog is on the chair,

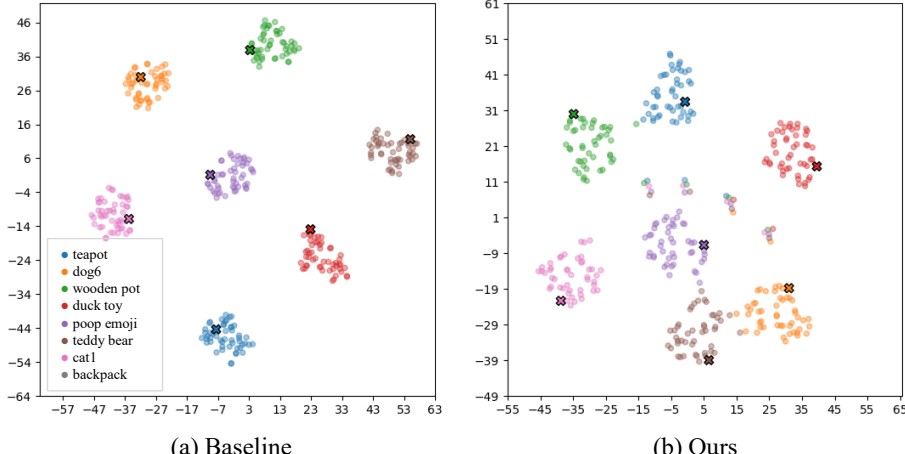

(a) Baseline          (b) Ours

Figure D: **Visualization of the prompt embedding distribution.** Each point denotes an embedding of a prompt that combines the personal concept token, *e.g., "[v]"*, with different context words *e.g., "a [v] dog at Eiffel Tower"*. The marker 'X' denotes the prompt without context *e.g., "a [v] dog."* As shown, the baseline yields a concentrated distribution around the prompt without contextual words, whereas our method yields a more dispersed distribution. The concentration in baseline results reflects representation collapse, where embeddings remain similar despite changes in contextual words. In contrast, our method effectively mitigates this collapse, thereby enhancing textual representation.

not on the right. For each prompt, we also provide its comparison to a non-personalized model, *i.e., prior.*, showing that the errors stem from this non-personalized-based model.

**Additional Qualitative Ablation Study.** We provide additional qualitative comparisons to visualize the impact of the proposed MLM on image generation in Figure E. Incorporating the proposed MLM during customization reduces suppression of contextual tokens, leading to their inclusion in the image or reinforcing associated attributes.

**Additional Cross-Attention Visualization.** We provide additional cross-attention visualization to visualize the impact of the proposed approach in Figures G and H.

**Additional Qualitative Comparison with Baselines.** We provide additional qualitative results of our approach in Figures I to L.

## C  ADDITIONAL ANALYSIS

### C.1  INTERPRETABLE EVIDENCE OF ENHANCEMENT IN TEXT SPACE

We provide additional interpretable evidence demonstrating the effectiveness of the proposed method. Specifically, we visualize the prompt embeddings derived from prompts including personal concept and different contextual words *e.g., "a [v] dog at Eiffel Tower."*. We project these embeddings into 2D space using UMAP (McInnes et al., 2018) and compare the results between the baseline and our method (Figure D). The analysis is conducted using 8 different concepts. As shown in Figure D (a), the prompt embeddings obtained from the baseline form a highly concentrated distribution around the prompt without contextual words *e.g., "a [v] dog"*. This concentration reflects textual representation collapse, where the semantics of the prompts remain similar despite variations in contextual words. In contrast, our method trained with MLM yields a dispersed prompt distribution (Figure D (b)), indicating that the representation collapse has been effectively addressed, and the representations are enhanced. Furthermore, we validate this observation by quantifying the overall concentrations of prompt embeddings in Table A. Specifically, we measure the average pair-wise cosine distance within each concept's prompt-embedding set. The results show that the baseline

| Method | Pair-wise Dist. ↑ |
|---|---|
| Baseline | 0.1268 |
| Ours | **0.2784** |

Table A: **Analysis of prompt embedding distribution.** Average pair-wise cosine distances are measured within each concept's prompt-embedding set. The baseline exhibits substantially lower distances compared to our method, indicating a highly concentrated distribution with collapse. The results are consistent with the visualization shown in Figure D.

exhibits substantially lower distances compared to our method with MLM, further confirming that it forms a collapsed embedding distribution, which is consistent with the visualization denoted in Figure D.

### C.2 IMPACT OF TEXTUAL ENHANCEMENT TO CROSS-ATTENTIONS

In Figure 4, cross-attention maps are visualized, where baseline results with collapsed textual representation lead to dispersed activations with low T2I alignment. We have validated and confirmed this observation in Table 1 of Section 2.3, where the baseline yields contextual tokens to become highly similar to the personal concept (lower $d_\text{text}$), and cross-attention maps to be dispersedly distributed (higher $\text{SV}_\text{cross}$). Here, we provide further explanation on *why* the textual collapse leads to dispersion in cross-attentions.

**Norms of personal concept tokens remain constant.** We first would like to note that the norms of the personal concept tokens remain constant after personalization. This constant trend has been reported in (Pang et al., 2024); however, we have further validated this in our own setting. Specifically, we compared the norms of personal concept tokens and contextual tokens in the encoded prompts for both the baseline and our method. We measured their ratio and found it to be approximately 1.005 in both methods, indicating that the norms of personal and contextual tokens are mostly identical.

**Collapsed text embeddings lead to similar cross-attention scores.** We explain why the similarity in text tokens impacts the cross-attention maps distribution. Let $\mathbf{Q} \in \mathbb{R}^{L_1 \times d}$ be the input image, and $\mathbf{K} \in \mathbb{R}^{L_2 \times d}$ be the input text tokens to cross attention layers in UNet, where $L_1$ denotes the number of image tokens and $L_2$ denotes the number of text tokens. The correlation $\mathbf{M} \in \mathbb{R}^{L_1 \times L_2}$ between image and text is computed as $\mathbf{M} = \mathbf{Q}\mathbf{K}^\top$. We would like to note that, when the cosine similarity between token embeddings become similar, *i.e.,* $\mathbf{K}[:, i] \approx \mathbf{K}[:, j]$ for $i \neq j$, where $i$ denotes the $i$-th the token embedding, $\mathbf{K}[:, i] \in \mathbb{R}^d$, the corresponding correlation also become similar, *i.e.,* $\mathbf{M}[:, i] \approx \mathbf{M}[:, j]$. Note that, since the norms of text tokens are mostly identical, the similarity between the correlations $\mathbf{M}[:, i]$ and $\mathbf{M}[:, j]$ are determined by $\cos(\mathbf{K}[:, i], \mathbf{K}[:, j])$. Intuitively, if all input text tokens are identically similar, then the resulting attention $\mathbf{A} \in \mathbb{R}^{L_1 \times L_2}$,

$$\mathbf{A} = \text{Softmax}(\mathbf{Q}/\sqrt{d}),$$

becomes uniform, as the $\mathbf{M}[k, :]$ becomes a evenly valued vector for all pixel $k$'s in the image. Note that the softmax is computed per-pixel, and the input tokens are competing at each pixel location. Therefore, textual representation affects how the cross-attentions are distributed, and collapsed representation leads to a dispersed cross-attention map distribution. Notably, we have validated in Table 1 that MLM effectively mitigates textual collapse with higher $d_\text{text}$, which leads to more concentrated and meaningful cross-attention estimates with lower $\text{SV}_\text{cross}$, and in turn improves generalization in image generation.

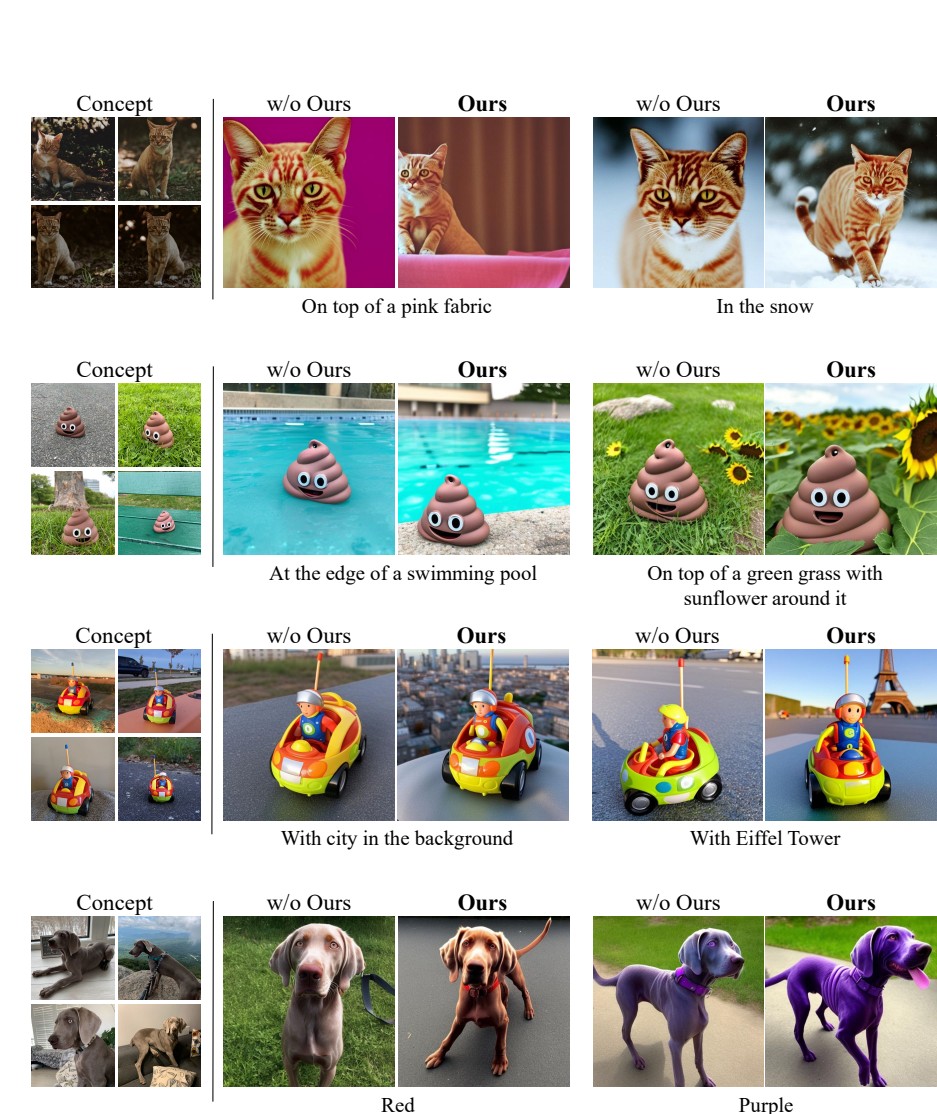

Figure E: **Additional Qualitative Ablation Study.**

1. a photo of a {} with the Taj Mahal in the background
2. a photo of a {} with the Amazon rainforest in the background
3. a photo of a {} with the Antarctic ice field in the background
4. a photo of a {} with the Caribbean beach in the background
5. a photo of a {} with the Egyptian pyramid in the background
6. a photo of a {} with the Eiffel Tower in the background
7. a photo of a {} with the Grand Canyon in the background
8. a photo of a {} with the Great Wall of China in the background
9. a photo of a {} with the Hollywood movie set in the background
10. a photo of a {} with the Mount Everest base in the background
11. a photo of a {} with the North Pole in the background
12. a photo of a {} with the Renaissance chapel in the background
13. a photo of a {} with the Roman Colosseum in the background
14. a photo of a {} with the Sahara Desert in the background
15. a photo of a {} with the San Francisco cable car in the background
16. a photo of a {} with the Times Square in the background
17. a photo of a {} with the Tokyo skyline in the background
18. a photo of a {} with the Venetian canal in the background
19. a photo of a {} with the ancient Egyptian tomb in the background
20. a photo of a {} with the ancient Greek agora in the background
21. a photo of a {} with the botanic garden in the background
22. a photo of a {} with the coral reef in the background
23. a photo of a {} with the craft brewery in the background
24. a photo of a {} with the crumbling ancient ruins in the background
25. a photo of a {} with the cutting-edge Silicon Valley startup in the background
26. a photo of a {} with the electric rock concert in the background
27. a photo of a {} with the energetic Brazilian carnival in the background
28. a photo of a {} with the enigmatic Stonehenge in the background
29. a photo of a {} with the fairy tale castle in the background
30. a photo of a {} with the industrial Detroit factory in the background
31. a photo of a {} with the industrial manufacturing zone in the background
32. a photo of a {} with the international space station in the background
33. a photo of a {} with the legendary Wild West saloon in the background
34. a photo of a {} with the lively street market in the background
35. a photo of a {} with the lush vineyard in the background
36. a photo of a {} with the majestic European cathedral in the background
37. a photo of a {} with the major city public library in the background
38. a photo of a {} with the medieval castle in the background
39. a photo of a {} with the modern art museum in the background
40. a photo of a {} with the mythical underwater Atlantis in the background
41. a photo of a {} with the old European cobblestone street in the background
42. a photo of a {} with the refined art gallery in the background
43. a photo of a {} with the secluded Himalayan monastery in the background
44. a photo of a {} with the serene city park in the background
45. a photo of a {} with the sleek modern office in the background
46. a photo of a {} with the spacious old warehouse in the background
47. a photo of a {} with the steep mountain pass in the background
48. a photo of a {} with the thick deep forest in the background
49. a photo of a {} with the urban chic rooftop in the background
50. a photo of a {} with the utopian space colony in the background
51. a photo of a {} with a screwdriver
52. a photo of a {} in front of a laptop
53. a photo of a {} beside a stack of books
54. a photo of a {} with a coffee cup
55. a photo of a {} with sunglasses
56. a photo of a {} by a window
57. a photo of a {} near a bicycle
58. a photo of a {} with a smartphone
59. a photo of a {} on a chair
60. a photo of a {} next to a painting
61. a photo of a {} in a garden
62. a photo of a {} beside a piano
63. a photo of a {} with a newspaper
64. a photo of a {} by a car
65. a photo of a {} with a backpack
66. a photo of a {} in front of a door
67. a photo of a {} beside a globe
68. a photo of a {} with a notebook
69. a photo of a {} on a sofa
70. a photo of a {} near a fountain
71. a photo of a {} with a pen
72. a photo of a {} with headphones
73. a photo of a {} in front of a brick wall
74. a photo of a {} beside a guitar
75. a photo of a {} with a water bottle
76. a photo of a {} at a bus stop
77. a photo of a {} near a bookshelf
78. a photo of a {} with a camera
79. a photo of a {} on the floor
80. a photo of a {} with a suitcase
81. a photo of a {} near flowers
82. a photo of a {} beside a whiteboard
83. a photo of a {} with an umbrella
84. a photo of a {} at a desk
85. a photo of a {} on a staircase
86. a photo of a {} near a street sign
87. a photo of a {} with a tablet
88. a photo of a {} beside a chalkboard
89. a photo of a {} with a dog
90. a photo of a {} by a tree
91. a photo of a {} with a hat
92. a photo of a {} with a cup of tea
93. a photo of a {} on a bridge
94. a photo of a {} beside a mirror
95. a photo of a {} in a park
96. a photo of a {} with a cat
97. a photo of a {} beside a lamp
98. a photo of a {} by the ocean
99. a photo of a {} near a campfire
100. a photo of a {} against a wall

Figure F: **The list of 100 prompts used for analytical experiments in Section 2.3.**

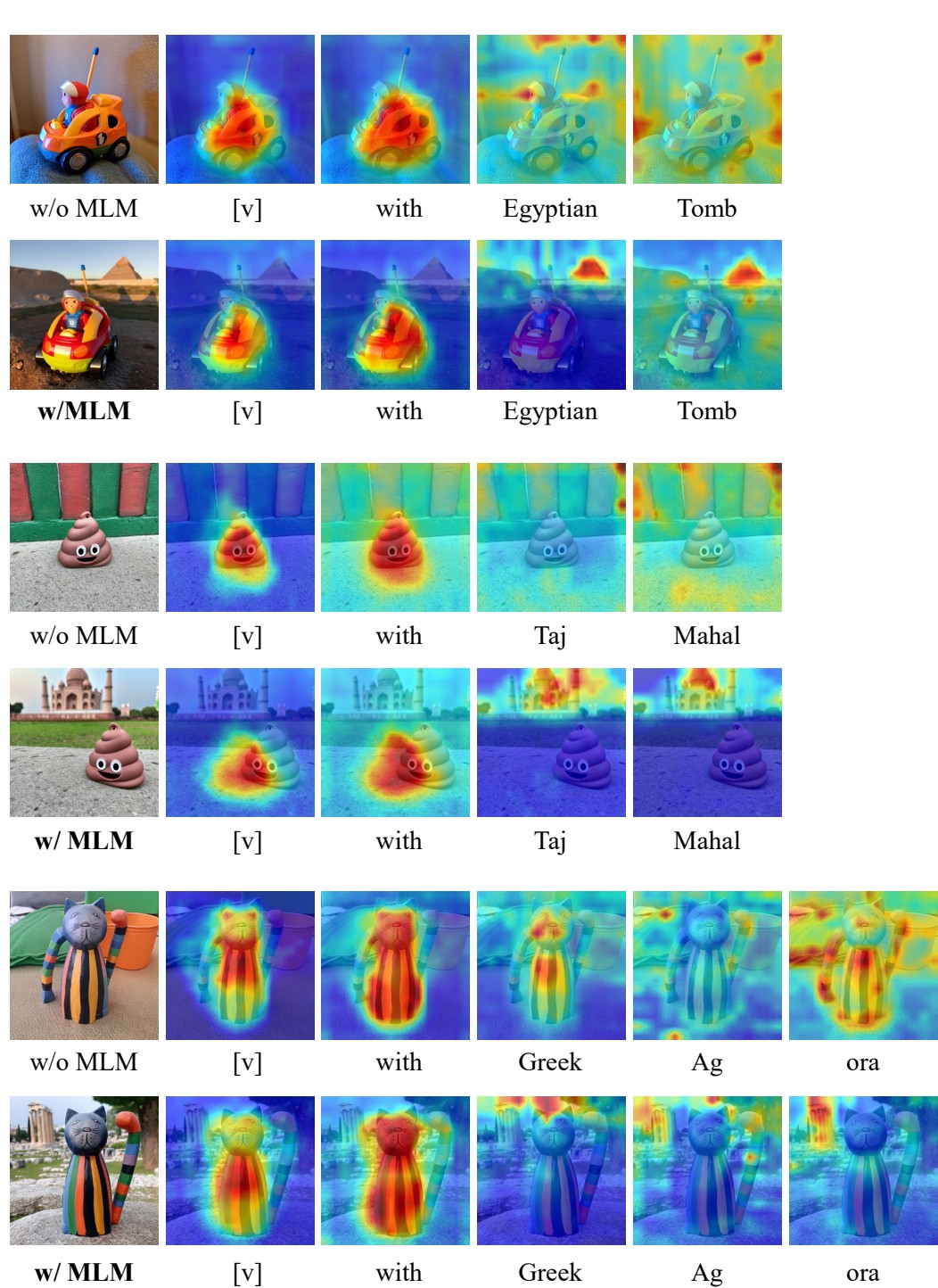

Figure G: **Additional Cross Attention Visualizations.**

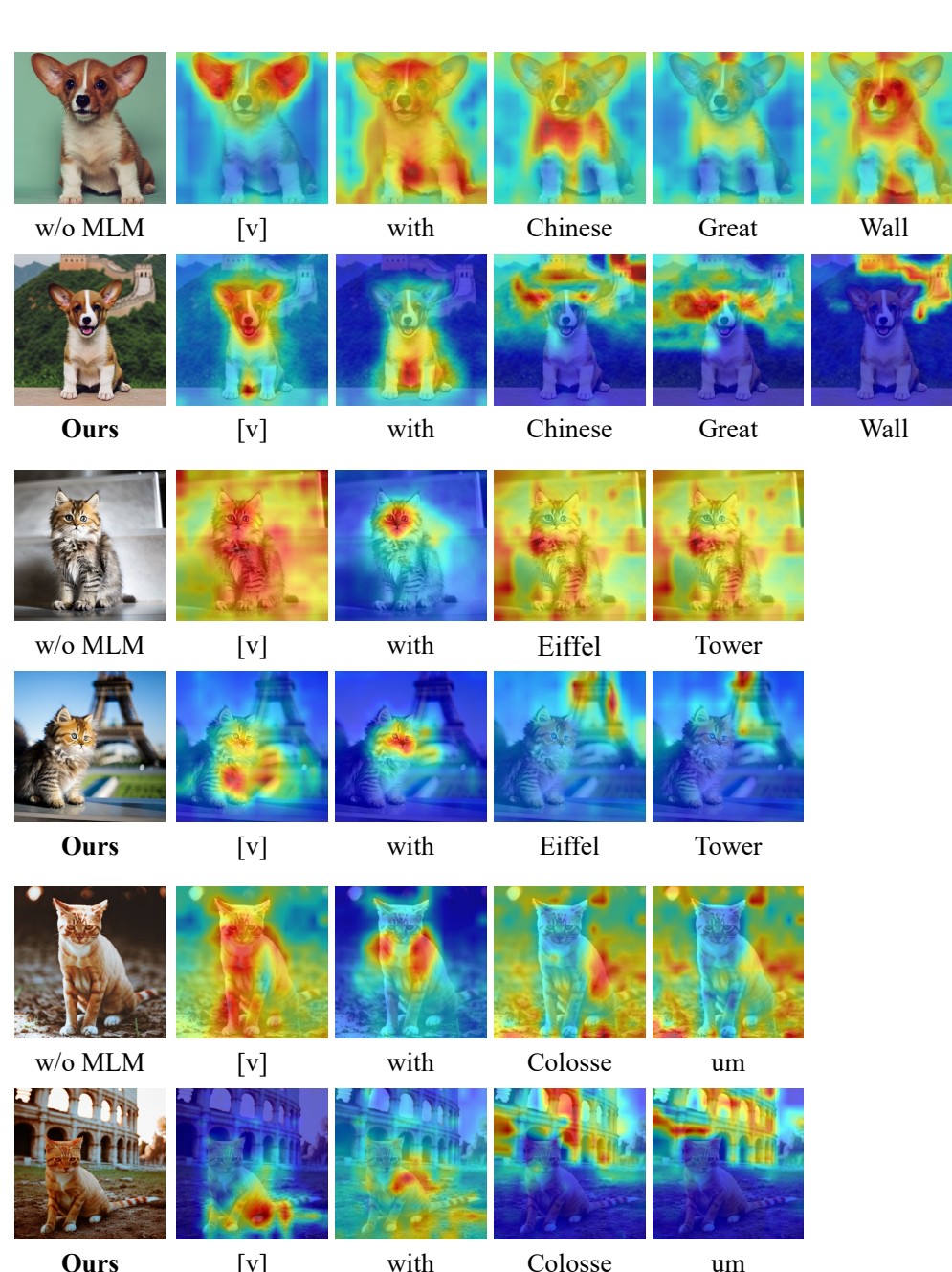

Figure H: **Additional Cross Attention Visualizations.**

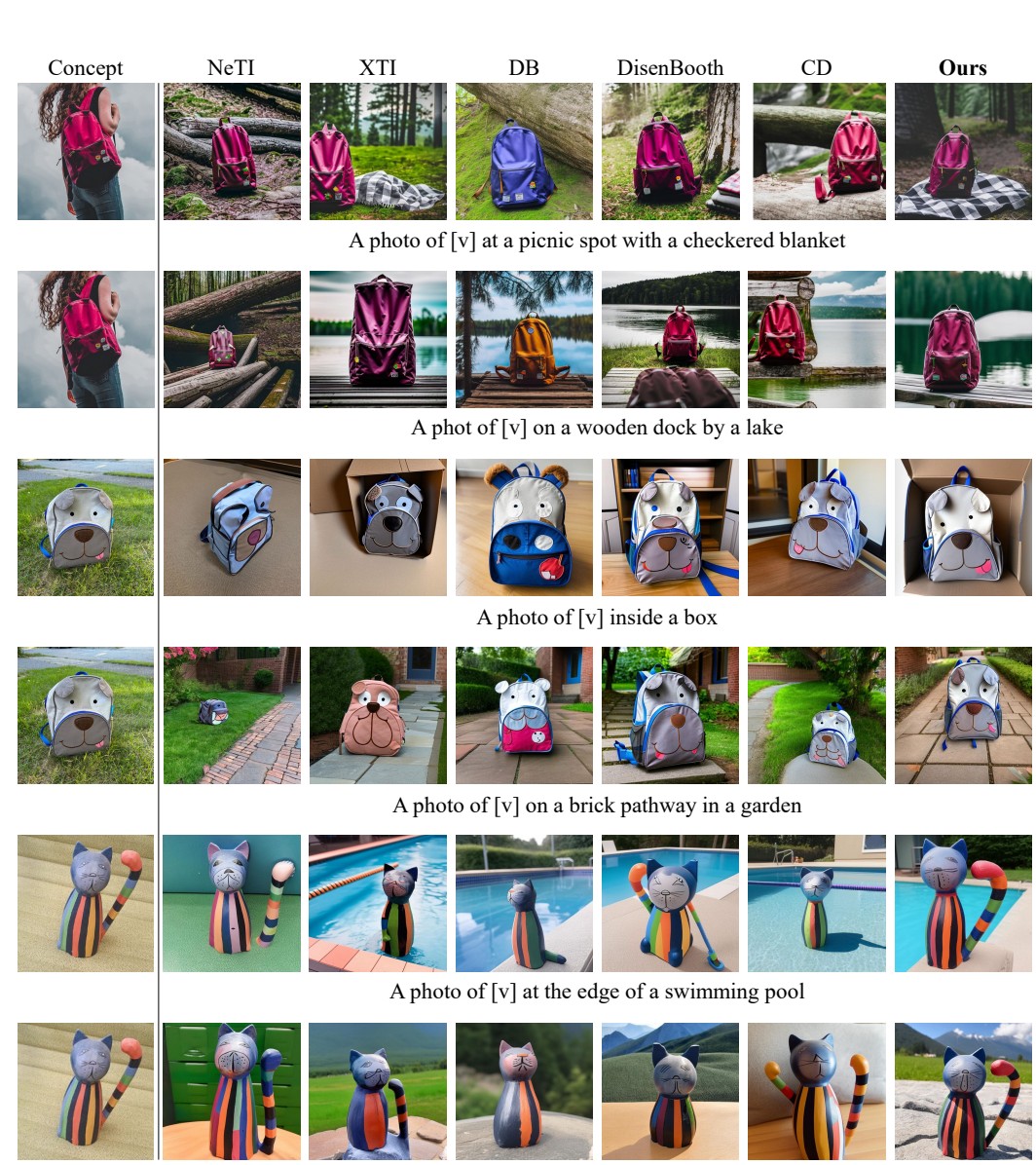

Figure I: **Additional qualitative comparisons results.**

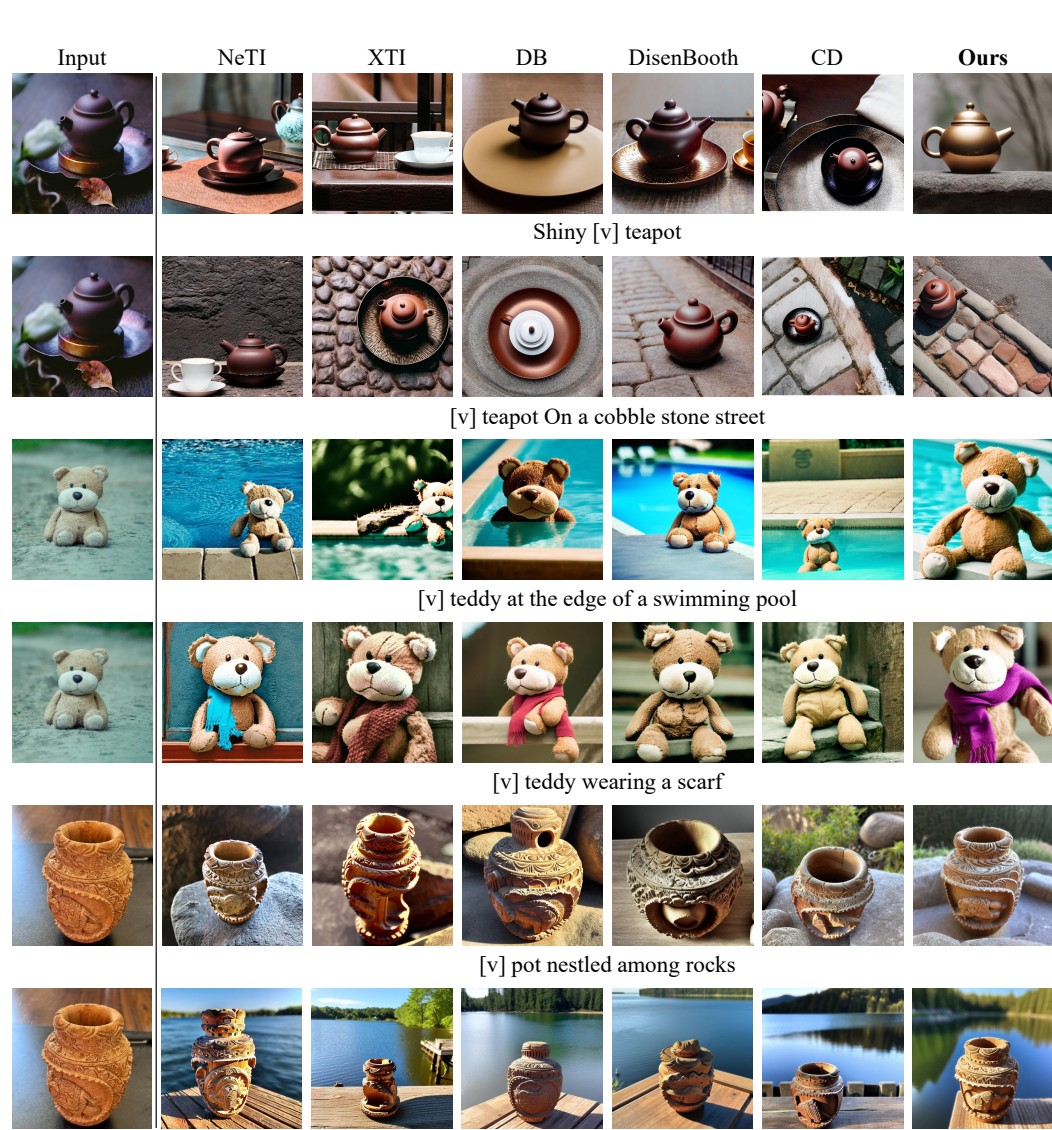

Figure J: **Additional qualitative comparisons results.**

1404
1405
1406
1407
1408
1409
1410
1411
1412
1413
1414
1415
1416
1417
1418
1419
1420
1421
1422
1423
1424
1425
1426
1427
1428
1429
1430
1431
1432
1433
1434
1435
1436
1437
1438
1439
1440
1441
1442
1443
1444
1445
1446
1447
1448
1449
1450
1451
1452
1453
1454
1455
1456
1457

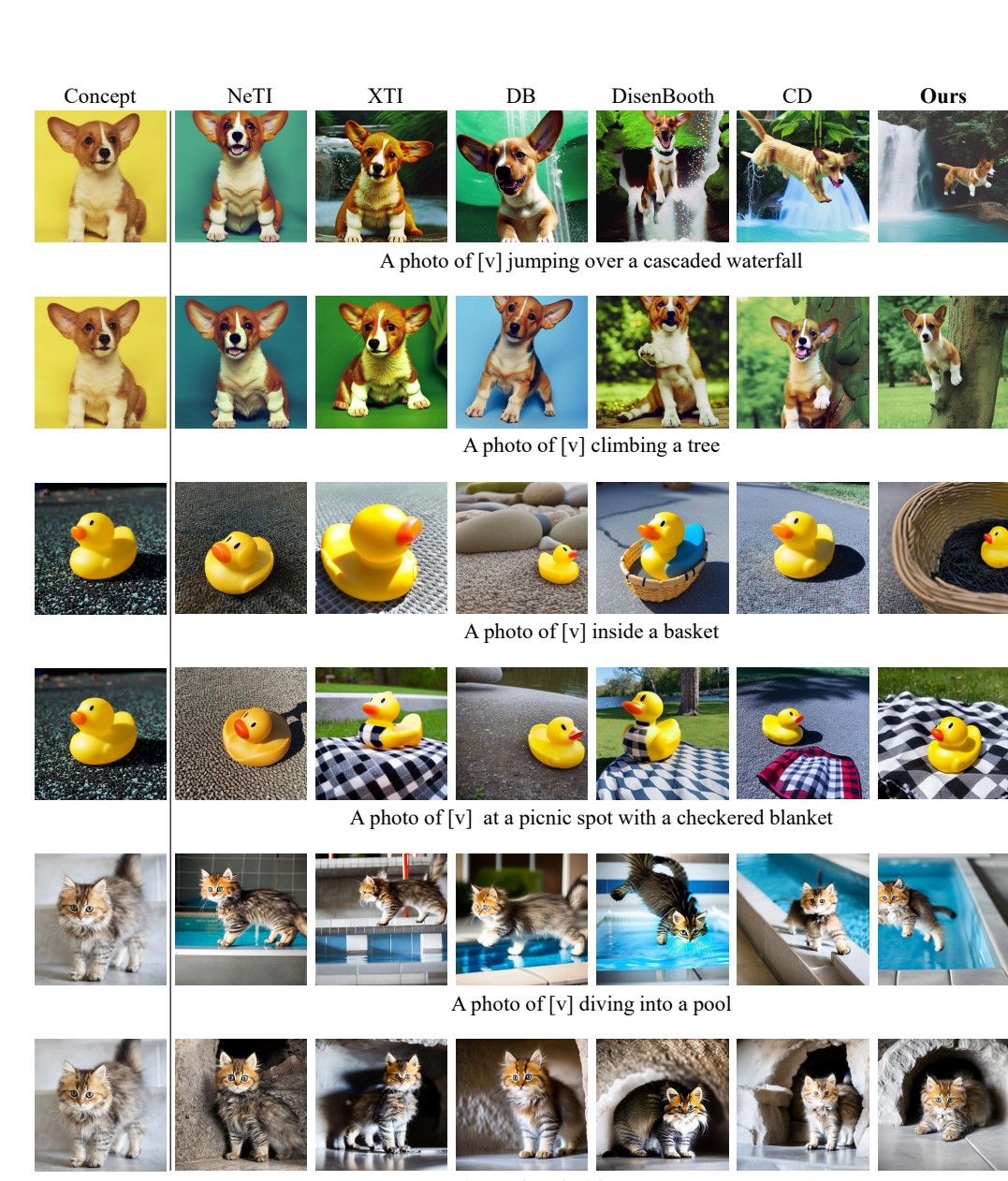

Figure K: **Additional qualitative comparison results.**

1458
1459
1460
1461
1462
1463
1464
1465
1466
1467
1468
1469
1470
1471
1472
1473
1474
1475
1476
1477
1478
1479
1480
1481
1482
1483
1484
1485
1486
1487
1488
1489
1490
1491
1492
1493
1494
1495
1496
1497
1498
1499
1500
1501
1502
1503
1504
1505
1506
1507
1508
1509
1510
1511

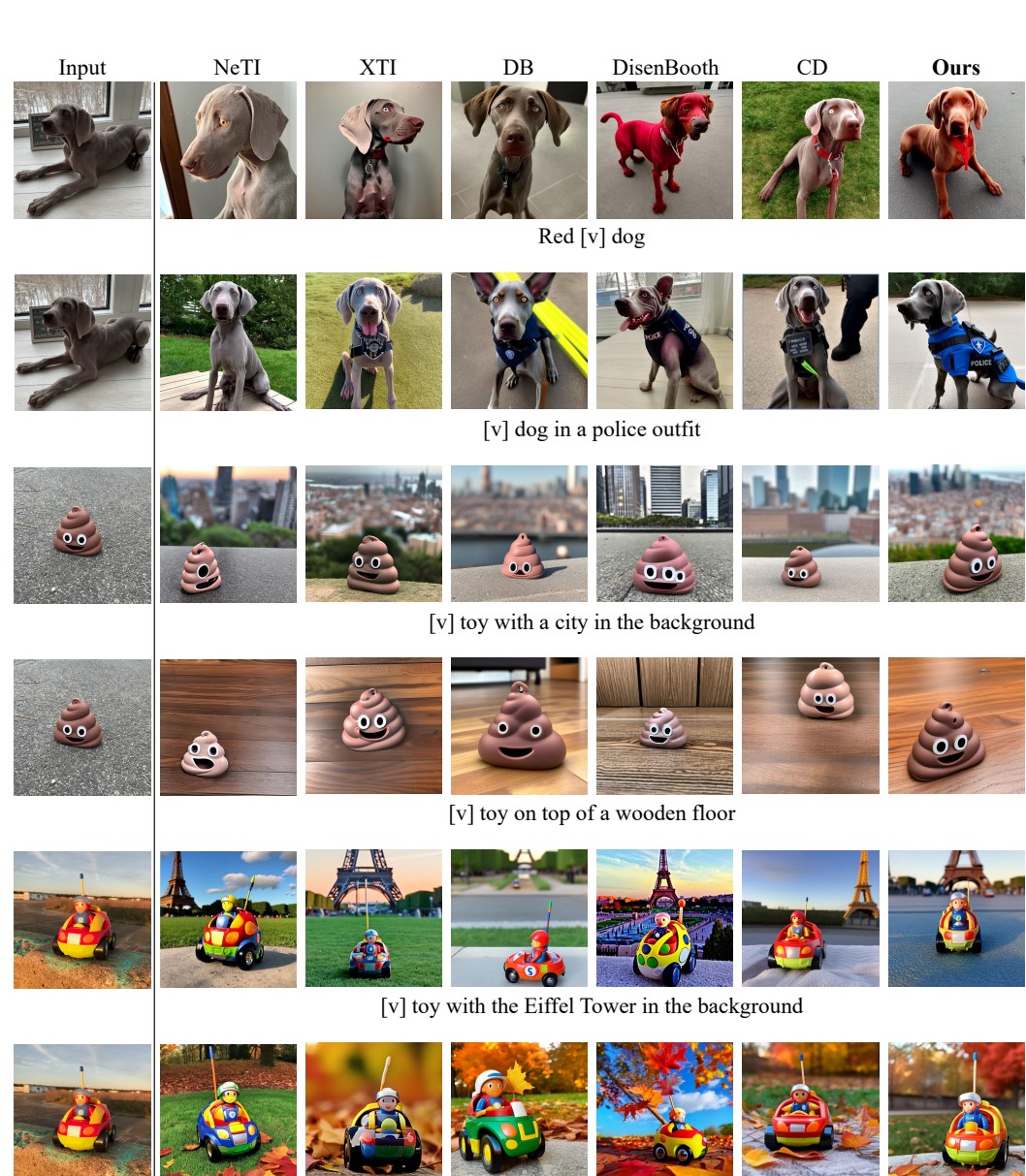

Figure L: **Additional qualitative comparisons results.**

