# OpenReview forum: "Learning Diverse Textual Contexts for Robust Personalization of Text-to-Image Diffusion Models"
_ICLR.cc/2026/Conference — ICLR 2026 Conference Withdrawn Submission_

### Official Review · Reviewer_TZWq · 2025-10-27

**Soundness:** 2
**Presentation:** 2
**Contribution:** 2
**Rating:** 2
**Confidence:** 5

**Summary:**

This paper focuses on the personalization task in the field of image generation. The authors proposed a method to enrich the training sample diversity by adopting MLM and a tuning-based personalization method. The experiment show the method can achieve competitive performance for this task.

**Strengths:**

- The presentation of figures is great and easy to understand.
- The math notations in this paper are self-contained and well-defined.
- The paper writing is easy to follow.

**Weaknesses:**

- I have to say that tuning-based personalization (like, DreamBooth, Custom Diffusion) is outdated. Learning-based personalization is the mainstream in the current image generation community.
- The key part of the proposed method is adopting MLM for Learning Diverse Contexts. However, the reason why MLM works is not straightforward and requires further explanation.
- The authors kept highlighting that the output token is the linear combination of the input tokens in MLM. But how is this related to personalization?
- The proposed method requires training a lightweight transformer network to bridge MLM and the CLIP text space. Why train another module for personalization fine-tuning? It's just so weird.
- The experiment in Fig. 4 is unreasonable. Why could the textual diversity affect the attention association?
- The compared methods are also outdated.

**Questions:**

Please see the section of weakness.

---

> ### Author Response · Authors · 2025-11-22
> **Response to Reviewer TZWq (1/3)**
>
> We sincerely appreciate your thoughtful comments. We hope that our responses below will adequately address your concerns.
>
> ---
> **W1. Tuning-based Method vs Learning-based Method.**
> > *I have to say that tuning-based personalization (like, DreamBooth, Custom Diffusion) is outdated. Learning-based personalization is the mainstream in the current image generation community.*
>
> We acknowledge that our method involves finetuning; however, current learning-based approaches face notable limitations.
> * **Inferior subject fidelity:** We observe that existing learning based approaches yield inferior results, especially in preserving the identity of subjects. To show this, we compare the existing learning based approaches:
>
> | **Method**           | **CLIP ↑** |  **ImageReward ↑** |**DINO ↑** |
> |------------------|------|------|--------|
> | ELITE         [1]|0.734 | 0.069  |0.693|
> | BLIP-D        [2]|0.745 | 0.206  |0.687|
> | IPAdapter     [3]|0.736 | 0.217  |0.699|
> | DisEnvisioner [4]|0.772  |0.610  |0.670|
> | Personalize Anything [5]| 0.711   | 0.711 |
> | Ours             |**0.785** |  **0.842**|**0.760**|
>
> The above table shows that a notable gap exists between our approach and recent learning based approaches. While DisEnvisioner achieves a comparable CLIP score for semantic alignment, this comes with a cost of subject fidelity, showing a substantially lower DINO score (-0.09). The qualitative comparisons for [1,2] are also available in Figure 6 in our main paper.
>
>
> * **Requirement of image collection in diverse contexts:** We also note that these learning-based approaches are based on a collection of images captured in diverse contexts, which is costly and often impractical to obtain. In contrast, our results are based on a small number (4-5) of images captured in limited contexts, yet still deliver superior generalization to novel prompts for generation, as reflected by higher CLIP and ImageReward scores.
>
>
>
>
>
>
>
>
>
>
>
> ---
> **W2. Why MLM works is not straightforward and requires further explanation.**
> > *The key part of the proposed method is adopting MLM for Learning Diverse Contexts. However, the reason why MLM works is not straightforward and requires further explanation.*
>
> The purpose of deploying MLM during personalization is to avoid the requirement of collecting images to be paired with diversified text prompts. We clarify below why simply diversifying text prompts cannot be a solution and how we address this with MLM.
>
> * **Challenge with diversified text prompts:** While diversifying text prompts by combining the personal concept token (e.g., [v]) with diverse contextual words, e.g., "[v] at Eiffel Tower", can be viewed as a simple process, incorporating these diversified prompts into T2I personalization is challenging. This is because T2I personalization typically requires an image that matches the input prompt as the reconstruction target. Therefore, a diversified text prompt alone cannot be utilized for personalization, and our key novelty lies in circumventing this restriction.
>
> * **MLM as circumventions:** Adopting MLM during personalization effectively addresses these requirements, as it operates entirely within the text space. In MLM, the model learns to predict a masked token by using contextualized representations produced through attention, where each output token is computed as a weighted linear combination of the input tokens. By including the personal concept token during MLM, the personal concept is repeatedly exposed to diverse semantics, and the model learns to align the concept so that the concept becomes semantically coherent with surrounding contexts. Note that, as MLM only requires an input text prompt and the masked token label for training, the diversified context learning can be effectively achieved without having paired images.

---

> ### Author Response · Authors · 2025-11-22
> **Response to Reviewer TZWq (2/3)**
>
> **W3. How is the linear combination of input tokens related to personalization?**
> > *The authors kept highlighting that the output token is the linear combination of the input tokens in MLM. But how is this related to personalization?*
>
>
> We highlighted linear combination as it explains how the personal concept token, e.g., [v], is contextualized with diverse semantics in text space. While the personal concept can be contextualized in image space (i.e., capturing images of personal concepts in diverse scenes), we instead contextualize the personal concept with diverse contexts in textual semantic space through linear combination. We clarify Equation (4) of the paper:
>
>
> $$
> \\begin{aligned}
> \\underbrace{\\textbf{O}[m,:]}\_{\\text{OutputMask}} &= \\sum\_{j=1}^{L} \\textbf{A}[m,j]\\textbf{V}[j,:] \\\\
> &= \\sum\_{\\substack{j=1 \\\\ j\\neq \*}}^{L} \\textbf{A}[m,j] \\underbrace{\\textbf{V}[j,:]}\_{\\text{Context}} + \\textbf{A}[m,\*] \\underbrace{\\textbf{V}[\*,:]}\_{\\text{Personal Concept}} ~,
> \\end{aligned}
> $$
>
>
> Where $\\mathbf{A}\\in\\mathbb{R}^{L\\times L}$ is a self-attention map, $\\mathbf{V}\\in\\mathbb{R}^{L\\times d}$ is the input tokens, and $\\mathbf{O}\\in\\mathbb{R}^{L\\times d}$ is the output tokes. The $j$ denotes the index of the $j$-th token $\\mathbf{V}[j,:]\\in\\mathbb{R}^{d}$, and $m$ and $*$ denotes the index of mask and personal concept. Utilizing the contextualized output mask and the MLM head $\\psi$, the MLM loss is computed,
>
> $$
> \\begin{align}
> \\begin{split}
> \\hat{\\mathbf{y}}&= \\psi(\\textbf{O}[m,:]), \\\\
>     \\mathcal{L}_\\text{MLM}&= \\mathbb{E}[\\text{CrossEntropy}(\\textbf{y},\\hat{\\mathbf{y}})].
>     \\end{split}
> \\end{align}
> $$
>
>
> As shown, the personal concept contributes to computing the MLM loss as it is combined in the output mask $\\mathbf{O}[m,:]$. Therefore, minimizing the loss encourages the personal concept embedding to be aligned so that it is semantically compatible with surrounding contexts. In this regard, we highlighted the linear combination because it explains how the personal concept is contextualized with diverse contexts in text space, and how it contributes to the MLM objective.
>
>
>
>
> ---
> **W4. The necessity of MLM head Pretraining.**
> > *The proposed method requires training a lightweight transformer network to bridge MLM and the CLIP text space. Why train another module for personalization fine-tuning?*
>
> We first would like to clarify that the MLM head pretraining is only done **once** and is shared by all the concept personalization. We argue that the pretraining is necessary, as the MLM head with random initialization provides noisy signals during initial training steps. To validate this, we compare a pretrained MLM head with a randomly initialized head that is simultaneously trained during personalization:
>
> | **MLM Head**           | **ImageReward ↑** | **CLIP ↑** |
> |------------------  |------    |------|
> | Baseline    (w/o MLM)|0.562     |0.774 |
> | Random Init.(w/ MLM) |0.579     |0.780 |
> | Pretrain    (w/ MLM) |**0.781** | **0.787**|
>
> As shown, the random initialization still leads to improvements from baseline (+0.017 ImageReward); however, it exhibits a notable gap compared to the head with pretraining (-0.202 ImageReward). This is primarily due to noisy signals provided during the initial training steps, which can be inaccurate. Since personalization is done with a small number of training steps, this noisy signal during the initial steps can be highly influential. Thus, a pretrained MLM head is needed to provide reliable guidance for personalization.

---

> ### Author Response · Authors · 2025-11-22
> **Response to Reviewer TZWq (3/3)**
>
> **W5. Why could textual diversity affect the cross-attention?**
> > *The experiment in Fig. 4 is unreasonable. Why could the textual diversity affect the attention association?*
>
> We would like to clarify that the proposed approach enhances the textual representation, and this effect further affects the cross-attention, as UNet accepts the text embeddings as input when computing cross-attention.
>
> * **Textual representation enhancement:** In Table 1 of Section 2.3, we first validated that textual representation collapses in baseline, where the token semantics become highly similar to the personal concept, as reflected in substantially lower $d_\text{text}$, which measures the cosine distance between personal concept and the context tokens. In contrast, by deploying MLM with diverse semantics, this is effectively addressed, showing notably greater $d_\text{text}$, indicating the personal concept and context tokens are distinct from each other without collapse.
>
> * **Impact on cross-attentions:** Also, in Figure 4, we visualized how textual representation affects the cross attention, where baseline results with collapsed textual representation lead to dispersed activations. This is primarily because cross-attention scores are computed based on the dot product of the input image and text tokens, and we would like to formally illustrate this. Let $\textbf{Q}\in\mathbb{R}^{L_1\times d}$ be the input image, and $\textbf{K}\in\mathbb{R}^{L_2\times d}$ be the input text tokens of cross attention layers in UNet. The correlation $\textbf{M}\in\mathbb{R}^{L_1\times L_2}$ between image and text is computed as,
> $$
> \textbf{M}=\textbf{Q}\textbf{K}^\top.
> $$
> When input tokens become similar, i.e., $\textbf{K}[i,:]\approx \textbf{K}[j,:]$ for all $i\neq j$, where $i$ and $j$ denotes the index of the token embedding $\textbf{K}[i,:]\in\mathbb{R}^{d}$, the corresponding correlation also become similar, i.e., $\textbf{M}[:,i]\approx \textbf{M}[:,j]$. By intuition, if all input text tokens are identical, then the resulting attention $\textbf{A}\in\mathbb{R}^{L_1\times L_2}$,
> $$
> \textbf{A}=\text{Softmax}(\textbf{M}/\sqrt{d}),
> $$
> becomes uniform, as the $\textbf{M}[k,:]$ becomes a evenly valued vector for all pixel $k$'s in the image. Therefore, textual representation affects how the cross-attentions are distributed, and collapsed representation leads to a dispersedly activated cross-attention map. Notably, we showed validation results in Table 1 that MLM effectively mitigates textual collapse (higher $d_\text{text}$), leading to concentrated and meaningful estimation of cross-attentions (lower $SV_\text{cross}$), yielding improved generalization in image generation.
>
>
>
>
>
>
>
>
>
>
>
>
>
>
>
>
> ---
> **W6. Additional baseline comparisons.**
> > *The compared methods are also outdated.*
>
>
>
>
> To address your concern, we provide additional comparisons, including several recent state-of-the-art personalization methods:
> | **Method**           | **ImageReward ↑** | **DINO ↑** |
> |------------------|------|------|
> | DisEnvisioner    [4]    | 0.610   |0.670  |
> | Personalize Anything [5]| 0.711   | 0.711 |
> | AttentionDreamBooth [6] |  0.833  | 0.732 |
> | Ours                    |**0.842**|**0.760**|
>
> The result shows that our approach consistently outperforms contemporary baselines across key evaluation metrics, further validating the effectiveness of our method.
>
>
> [1] ELITE: Encoding Visual Concepts into Textual Embeddings for Customized Text-to-Image Generation, ICCV 2023.
> [2] BLIP-Diffusion: Pre-trained Subject Representation for Controllable Text-to-Image Generation and Editing, NeuRIPS 2023.
> [3] IP-Adapter: Text Compatible Image Prompt Adapter for Text-to-Image Diffusion Models, Arxiv 2023.
> [4] DisEnvisioner: Disentangled and Enriched Visual Prompt for Customized Image Generation, ICLR 2025.
> [5] Personalize Anything for Free with Diffusion Transformer, Arxiv 2025.
> [6] AttnDreamBooth: Towards Text-Aligned Personalized Text-to-Image Generation, NeurIPS 2024.

---

> ### Comment · Reviewer_YPra · 2025-11-26
>
> Thank you for providing additional experiments and further clarification. The new results address my concerns to some extent. The analysis regarding how textual semantic diversity improves generalization has been strengthened; however, the evidence still largely remains at the level of correlational observations. In addition, the methodological novelty appears to lie primarily in a conceptual re-formulation rather than substantial technical innovation. My overall assessment has improved, but I still lean slightly toward rejection.

---

> > ### Author Response · Authors · 2025-11-28
> > **Follow-Up Response to Reviewer YPra**
> >
> > We greatly appreciate Reviewer YPra's follow-up comments and active engagement in the discussion. We hope the response below helps further address the concerns.
> >
> > ---
> > * **Additional Interpretable Evidence.** In Figure D in Section C.1 in the revised Appendix, we have provided additional interpretable evidence that directly demonstrates the effectiveness of our approach.
> > Specifically, we have visualized the distributions of embeddings of prompts that combine personal concepts with various contextual words, e.g., "a [v] dog at Eiffel Tower," using UMAP. As shown, the prompt embeddings produced by the baseline method form a highly concentrated distribution around the prompt without contextual words, e.g., "a [v] dog", indicating representation collapse and loss of contextual semantics. In contrast, our method shows the opposite trend, yielding a more dispersed embedding distribution without such a collapse. We further validate this observation by measuring the pair-wise cosine distances of the prompt embeddings within each concept's embedding set:
> >
> > | Method            | **Pair-wise Distance ↓**|
> > |-------------------|----------------------|
> > | Baseline (w/o MLM)|0.127                 |
> > | Ours (w/ MLM)     |**0.278**             |
> >
> > As shown above, the baseline exhibits substantially lower average distance compared to our method trained with MLM, which suggests that the prompt embeddings are closely located to each other, and exhibit collapse. This result, along with visualization (Figure D), demonstrates the effectiveness of our approach in mitigating the representational collapse in textual space. As explained in **W2** of our previous response, this textual enhancement is critical, as this enhancement leads to accurate cross-attention estimation with improved generalization in image generation. We have added details on how textual enhancement influences cross-attention in Section C.2 of the revised Appendix.
> >
> >
> >
> > * **Additional Clarification on Novelty.** We would like to respectfully note that the novelty of our approach lies in different aspects than architectural innovation. Instead, our method focuses on **diversifying textual contexts** and **learning these diversified contexts in text space**, independent of architectural designs. To the best of our knowledge, no prior work has demonstrated the importance of textual context diversification or proposed a method for learning such diversified contexts for personalization without relying on additional images. In addition, as our method is not limited to a specific architecture, it can be easily integrated into existing architectures. To demonstrate this compatibility, we incorporate the proposed method into TI [1] and XTI [2] and present the results below.
> >
> >
> > | Method           | **CLIP ↑**| **ImageReward ↑**|
> > |------------------|------|------|
> > | TI (w/o MLM)          |0.682| -0.769|
> > | TI (w/ MLM)            |**0.692**|**-0.571** |
> > | XTI (w/o MLM)          |0.748| 0.414|
> > | XTI (w/ MLM)            |**0.768**|**0.601** |
> >
> > As shown, integrating our method into existing approaches consistently yields performance gains, highlighting both its effectiveness and compatibility.
> >
> >
> > [1] An Image is Worth One Word: Personalizing Text-to-Image Generation using Textual Inversion, ICLR 2023
> > [2] P+: Extended Textual Conditioning in Text-to-Image Generation, Arxiv 2023

---

> ### Author Response · Authors · 2025-12-04
> **Additional Clarification to Reviewer TZWq Regarding W5**
>
> We would like to provide a minor clarification regarding **W5** (justification for why textual enhancement affects cross-attention). We explained that the dot-product scores $\textbf{M}[:,i]=\textbf{Q}\cdot\textbf{K[i,:]}^\top$ and $\textbf{M}[:,j]=\textbf{Q}\cdot\textbf{K[j,:]}^\top$ become similar when corresponding token embeddings are similar, that is, when $\textbf{K[i,:]}\approx \textbf{K[j,:]}$ with a small cosine distance. We would like to clarify that this holds under the assumption that $||\textbf{K}[i,:]||$ and $||\textbf{K}[j,:]||$ are equal, which we assumed.
>
> Although prior work [7] has shown that the norms of personal concept tokens remain similar, we further validated this in our own setting. Specifically, we compared the norms of personal concept tokens and contextual tokens in the encoded prompts for both the baseline and our method. We measured their ratio and found it to be approximately 1.005 in both methods, indicating that the norms are mostly identical. While this does not change our original conclusion, it provides more rigorous justification.
>
> [7] Cross Initialization for Personalized Text-to-Image Generation, CVPR 2024

---

### Official Review · Reviewer_YPra · 2025-10-30

**Soundness:** 2
**Presentation:** 2
**Contribution:** 2
**Rating:** 4
**Confidence:** 3

**Summary:**

This paper tackles overfitting in personalized text-to-image generation caused by limited training images. It propose a context diversification strategy in the text space: by applying random masking and semantic replacement to expand the text prompts, the model is encouraged to learn consistent semantic attributes of the target concept under diverse contexts, thereby improving generalization. The method improves generation quality and diversity without extra images and is validated via CLIPScore, attention maps, and embedding similarity analysis.

**Strengths:**

•	The paper introduces a novel perspective by enhancing concept binding through diversified textual semantic contexts，represents a clear and valuable conceptual shift.

•	The proposed method is lightweight and practical, requiring no additional images or modifications to the architecture of the diffusion model.

•	The experimental analysis is interpretable, using semantic alignment and attention-based diagnostics to demonstrate why the method works.

**Weaknesses:**

•	The approach can be viewed as a textual data augmentation strategy. The novelty may appear less substantial given the limited architectural or algorithmic innovation.

•	The core hypothesis—that semantic diversity in text can serve as an effective regularization signal similar to image diversity—lacks rigorous validation. It would be more valuable if more interpretable evidence could be provided to directly establish the link between text diversity and improved generalization ability.

•	Comparisons with more recent and stronger personalization baselines (e.g., Custom Diffusion, ELITE, SVDiff) are missing, making it difficult to fully assess the competitiveness of the method.

•	The experiments are mainly limited to common, coarse object categories. It is still unclear whether this method can be extended to concepts with stronger identity constraints (for example, faces or fine-grained categories).

**Questions:**

•	Can the author provide more evidence to support the regularization effect on text semantic diversity?

•	How do the results compare with recently stronger baseline methods?

•	 The rest can be seen in the Weaknesses .

---

> ### Author Response · Authors · 2025-11-22
> **Response to Reviewer YPra (1/3)**
>
> We sincerely appreciate Reviewer YPra for thoughtful comments and suggestions in reviewing our work. We address the questions and concerns as follows.
>
>
>
> ---
> **W1. Novelty: Is the approach simply a textual data augmentation?**
> > *The approach can be viewed as a textual data augmentation strategy. The novelty may appear less substantial given the limited architectural or algorithmic innovation.*
>
>
> We would like to note that textual context augmentation is only a part of the contribution. In fact, the contributions are twofold: 1) **context diversification in text space** and 2) **context learning in text space**. Below, we explain why context diversification can be a non-trivial process, the challenge it introduces, and how our method resolves it.
>
>
> * **Challenge with context diversification in text space:**. While diversifying text prompts by combining the personal concept token (e.g., [v]) with diverse contextual words, e.g., "[v] at Eiffel Tower", can be viewed as a simple process, incorporating these diversified prompts into T2I personalization is challenging. This is because T2I personalization typically requires an image that matches the input prompt as the reconstruction target. To avoid this restriction, we propose to learn the context within text space, and our key novelty lies in these circumventions.
>
> * **MLM as circumventions:** In our method, we avoid the need for paired images by *learning contexts in text space*. To achieve this, we adopt MLM, which operates entirely within text space. In MLM, the model learns to predict a masked token by using contextualized representations produced through attention, where each output token is computed as a weighted linear combination of the input tokens. By including the personal concept token during MLM, the personal concept is repeatedly exposed to diverse semantics through these combinations, and the model learns to align the concept to become semantically coherent with surrounding contexts. Note that, as MLM only requires an input text prompt and the masked token label for training, the diversified context learning can be effectively achieved without having paired images.

---

> ### Author Response · Authors · 2025-11-22
> **Response to Reviewer YPra (2/3)**
>
> ---
> **W2. Interpretable evidence with validation establishing the link between text diversity and improved generalization.**
> > *The core hypothesis—that semantic diversity in text can serve as an effective regularization signal similar to image diversity—lacks rigorous validation. It would be more valuable if more interpretable evidence could be provided to directly establish the link between text diversity and improved generalization ability.*
>
> Thank you for pointing out the importance of providing more interpretable evidence connecting textual diversity to generalization. Our analytical study in Section 2.3 was designed to address this connection, and here we would like to provide clarification on how the enhancement in text space affects the image generation.
>
> * **Textual enhancement:** In Table 1 of Section 2.3, we validated that textual representation exhibits collapses in baseline, where the token semantics become highly similar to the personal concept, as reflected in substantially lower d$_\text{text}$, which measures the cosine distance between personal concept and the context tokens. In contrast, by deploying MLM with diverse semantics.
>
> * **Impact on cross-attentions:** As interpretable evidence, we visualized the cross attention in Figure 4, where baseline results with collapsed textual representation lead to dispersed activations. This is primarily because cross-attention scores are computed based on the dot product of the input image and text tokens, and we would like to formally illustrate this. Let $\textbf{Q}\in\mathbb{R}^{L_1\times d}$ be the input image, and $\textbf{K}\in\mathbb{R}^{L_2\times d}$ be the input text tokens of cross attention layers in UNet. The correlation $\textbf{M}\in\mathbb{R}^{L_1\times L_2}$ between image and text is computed as,
> $$
> \textbf{M}=\textbf{Q}\textbf{K}^\top.
> $$
> When input tokens become similar, i.e., $\textbf{K}[i,:]\approx \textbf{K}[j,:]$ for all $i\neq j$, where $i$ denotes the $i$-th the token embedding $\textbf{K}[i,:]\in\mathbb{R}^{d}$, the corresponding correlation also become similar, i.e., $\textbf{M}[:,i]\approx \textbf{M}[:,j]$. In extreme cases, if all input text tokens are identical, then the resulting attention $\textbf{A}\in\mathbb{R}^{L_1\times L_2}$,
> $$
> \textbf{A}=\text{Softmax}(\textbf{Q}/\sqrt{d}),
> $$
> becomes uniform, as the $\textbf{M}[k,:]$ is evenly valued for all pixel $k$'s in the image. Therefore, textual representation affects how the cross-attentions are distributed, and collapsed representation leads to a dispersedly activated cross-attention map. Notably, we showed in Table 1 that MLM effectively mitigates textual collapse with higher $d_\text{text}$, which leads to more concentrated and meaningful cross-attention estimates with lower $SV_\text{cross}$, and in turn improves generalization in image generation.
>
>
>
>
>
>
>
>
>
> ---
> **W3. Comparison with more recent and stronger personalization baselines is missing.**
> > *Comparisons with more recent and stronger personalization baselines (e.g., Custom Diffusion, ELITE, SVDiff) are missing, making it difficult to fully assess the competitiveness of the method.*
>
> We note that we have provided the comparison with Custom Diffusion (CD) and ELITE in Tables 2 and 3, where our method outperforms CD and ELITE on all metrics, CLIP, DINO, ImageReward, and BLIP-VQA score, on both DreamBench and the challenging benchmark. Below, we provide a comparison with SVDiff. As shown, our method also outperforms SVDiff on all evaluation metrics.
>
>
>
> | Method           | **CLIP ↑** | **DINO ↑** | **ImageReward ↑** | **BLIP-VQA ↑** |
> |------------------|------|------|--------|----------|
> | SVDiff           |0.771 | 0.720| 0.487  |0.641|
> | Ours             |**0.785** | **0.760**| **0.842**|**0.726**|

---

> ### Author Response · Authors · 2025-11-22
> **Response to Reviewer YPra (3/3)**
>
> ---
> **W4. Extension to fine-grained categories.**
> > *The experiments are mainly limited to common, coarse object categories. It is still unclear whether this method can be extended to concepts with stronger identity constraints (for example, faces or fine-grained categories).*
>
>
>
>
> Our method is not limited to coarse-grained categories and can be easily applied to fine-grained concepts. To demonstrate this, we conducted personalization experiments on face images and compared our results with a baseline without MLM:
>
>
>
>
> | Method           | **CLIP ↑** | **ImageReward ↑** |
> |------------------|------|------|
> | Baseline         |0.751 | 1.043|
> | Ours             |**0.762**|**1.106** |
>
> The result above shows that the effectiveness of the proposed approach is not limited to coarse-grained subjects. In addition, we included qualitative personalization results with various input prompts for face images in Figure B of the Appendix Section B.

---

> ### Author Response · Authors · 2025-12-04
> **Additional Clarification to Reviewer YPra on W2**
>
> We would like to provide a minor clarification regarding **W2** (justification for why textual enhancement affects cross-attention). We explained that the dot-product scores $\textbf{M}[:,i]=\textbf{Q}\cdot\textbf{K[i,:]}^\top$ and $\textbf{M}[:,j]=\textbf{Q}\cdot\textbf{K[j,:]}^\top$ become similar when corresponding token embeddings are similar, that is, when $\textbf{K[i,:]}\approx \textbf{K[j,:]}$ with a small cosine distance. We would like to clarify that this holds under the assumption that $||\textbf{K}[i,:]||$ and $||\textbf{K}[j,:]||$ are equal, which we assumed.
>
> Although prior work [1] has shown that the norms of personal concept tokens remain similar, we further validated this in our own setting. Specifically, we compared the norms of personal concept tokens and contextual tokens in the encoded prompts for both the baseline and our method. We measured their ratio and found it to be approximately 1.005 in both cases, indicating that the norms are mostly identical. While this does not change our original conclusion, it provides more rigorous justification.
>
> [1] Cross Initialization for Personalized Text-to-Image Generation, CVPR 2024

---

### Official Review · Reviewer_SYkC · 2025-10-31

**Soundness:** 3
**Presentation:** 4
**Contribution:** 3
**Rating:** 8
**Confidence:** 4

**Summary:**

This paper proposes a novel and cost-effective method to inject contextual diversity by leveraging the text space compared to how prior works that diversify cost via using an increased number of images. The key technical contribution is utilizing a Masked Language Modeling (MLM) objective on a large set of automatically generated, diverse text prompts (e.g., "object at Eiffel Tower") during personalization.

**Strengths:**

1. The most interesting part is how the authors attempt tackling one of the most practical issue faced (i.e. lack of diverse context in training images- usually there are only 3-5 reference images available for a concept). Since MLM operates entirely in the text space, it does not require paired images for these diverse contexts, overcoming the main limitation of having to collect/curate more data per concept.
2. The proposed method is sufficiently novel and cost-effective. Prior works have focussed quite a bit on different ways of customizing diffusion models, but this paper attempts to solve a common fundamental limitation of ensuring that the learned custom embedding does not overfit the provided reference images.
3. The qualitative results in Figure 6 demonstrate that the proposed method outperforms baselines significantly in terms of editability and context generation (e.g. purple dress for the customized cat in the second row)
4. The performance gains seen via ImageReward in Tables 4-6 further strengthen the claims of the proposed method

**Weaknesses:**

1. The quality/diversity of the training is dependent on the predefined templates/words used for creating the prompt set. It is unclear whether the different contexts generated during inference are limited by the ones generated by the LLM. A discussion on the sensitivity to the source or quality of the diverse prompts would strengthen this point.
2. In a couple examples, the baselines seem to have better preservation of the custom concept features. For example, in the first row in figure 6, the red color present near the hair and hands of the toy are missing in the proposed method's results- while those details are missing for baselines too, would appreciate any intuition from the authors on why the reconstruction may not be perfect.

**Questions:**

The main question I have is why it was necessary to add a separate transformer and whether adding a linear prediction head on top of the unmodified, frozen CLIP encoder, or finetuning the existing higher layers of the CLIP encoder with the MLM objective would have helped as well?

---

> ### Author Response · Authors · 2025-11-22
> **Response to Reviewer SYkC**
>
> We sincerely thank Reviewer SYkC for the positive assessment of our work with constructive and thoughtful comments. We would like to address each of your questions and comments in detail below.
>
>
> ---
> **W1. Discussion on the sensitivity of the source or quality of diverse prompts.**
> > *The quality/diversity of the training is dependent on the predefined templates/words used for creating the prompt set. It is unclear whether the different contexts generated during inference are limited by the ones generated by the LLM. A discussion on the sensitivity to the source or quality of the diverse prompts would strengthen this point.*
>
>
> We believe this is an important point as the potential bias in the prompt set may affect the result. To examine the sensitivity, we provide comparison results obtained with different MLM prompt sources.  Specifically, we provide comparison results obtained with the LLM-generated captions and human-written captions. For LLM-generated captions, we obtained 100 different prompts for each category by querying the LLM. For human-written captions, we adopted COCO captions and replaced the subject word that exists in the prompt with a personal concept token. We would also like to clarify that the prompts set that we used in our paper are template-based, where we combined different context words with templates rather than querying the LLM to generate full prompts.
> | **Prompt Source**     | **CLIP ↑**        |
> |----------------|-------------|
> | Baseline | 0.770 |
> | COCO | 0.788 |
> | LLM | 0.790 |
> | Template | 0.786 |
>
> All three prompt sources yield consistent improvements over the baseline without MLM. The differences between them are small, which shows that our approach is not overly sensitive to the specific choice of MLM prompt source. This also indicates that having diverse contextual semantics can be more critical than the specific source of the prompts.
>
>
>
>
>
>
>
>
>
> ---
> **W2. While the baseline approaches are also imperfect, why do the reconstruction details appear imperfect in some examples?**
> > *...while those details are missing for baselines too, would appreciate any intuition from the authors on why the reconstruction may not be perfect.*
>
>
>
> We believe a potential reason is the inherent behavior of the Stable Diffusion v2.1 backbone. SD models are well known to exhibit issues such as incorrect attribute binding and attribute leakage, where attributes from the prompt or the model’s prior inadvertently override or blend with subject-specific details.
> Since the T2I personalization model is initialized from this backbone, such behavior can still influence the result and limit the preservation of fine details. We would also like to refer the reviewer to Figure C in the revised paper, which shows failure cases commonly presented in the base model.
>
>
>
>
>
>
>
>
>
>
> ---
> **Q1. Could applying linear prediction on top of a partially fine-tuned CLIP encoder with MLM have helped?**
> > *The main question I have is why it was necessary to add a separate transformer and whether adding a linear prediction head on top of the unmodified, frozen CLIP encoder, or finetuning the existing higher layers of the CLIP encoder with the MLM objective would have helped as well?*
>
>
> Yes, fine-tuning the part of the CLIP text encoder with linear also works. To show this, we fine-tuned the top three layers of the CLIP text encoder with linear projection:
>
> | **FineTuning**     | **CLIP ↑**        |
> |----------------|-------------|
> | Top3+Linear (w/o MLM)| 0.693 |
> | Top3+Linear (w/ MLM) | **0.759**  |
>
> As shown, applying the MLM objective with linear projection layers consistently leads to improvement. This result indicates that the linear projection can still serve to bridge the CLIP and MLM space.

---

### Official Review · Reviewer_tM5B · 2025-11-01

**Soundness:** 4
**Presentation:** 3
**Contribution:** 3
**Rating:** 6
**Confidence:** 3

**Summary:**

The paper tackles robustness issues in few-shot text-to-image (T2I) personalization caused by limited contextual diversity of the user-provided examples. Instead of collecting more images, it proposes to diversify the textual contexts of the personalized token (e.g., “[v] at Eiffel Tower”, “[v] in Pop Art style”) and to learn these contexts entirely in text space via a masked language modeling (MLM) objective. Concretely, during DreamBooth+LoRA fine-tuning, an auxiliary MLM loss is computed on a large, curated prompt set where some tokens are masked and predicted by a lightweight transformer head plugged on top of the CLIP text encoder. The personal concept token is contextualized through self-attention as a linear combination of context and concept tokens, encouraging the concept embedding to carry richer semantics without requiring paired images.

**Strengths:**

1 Clear, practical idea: address contextual overfitting by diversifying prompts in text space and enforcing MLM consistency, avoiding costly image collection. The approach is orthogonal to, and easily composable with, common personalization pipelines (DreamBooth+LoRA).

2 Solid empirical gains: consistent SOTA across CLIP (text alignment), DINO (subject fidelity), ImageReward (human preference proxy), and BLIP-VQA on two benchmarks. Qualitative examples show better prompt adherence while preserving identity.

3 Insightful analyses: textual-space cosine distances and image-space cross-attention diagnostics convincingly link the MLM objective to reduced semantic collapse and improved grounding.

4 Implementation details and ablations: reports sensitivity to λ, masking probability, and prompt set size; excludes benchmark prompts from MLM training to avoid leakage; single-GPU feasibility.

**Weaknesses:**

1 Dependence on curated prompt sets and LLMs: The large, hand/LLM-crafted prompt bank (tens of thousands) is central to performance. The construction process, coverage, and potential bias are only partially formalized; generalization to other domains/languages remains unclear.

2 Extra component and training stage: Requires pretraining a separate MLM head (albeit lightweight) on COCO+constructed prompts. The integration details (which layers are updated, interaction with CLIP tokenization nuances) could be more rigorously specified. It also assumes MLM is beneficial while keeping CLIP frozen beyond LoRA, which may not always hold.

3 Metric reliance: CLIP, DINO, ImageReward, BLIP-VQA have known limitations. There is no human user study or identity retrieval evaluation to triangulate improvements, and no failure analysis by prompt category (e.g., heavily compositional, relational, or negations).

**Questions:**

1 Multiple personalized tokens and continual personalization: Can the approach handle multiple personal concepts jointly, or sequential personalization without interference?

2 Data leakage and safety: How do you ensure the MLM prompt set does not inadvertently include near-duplicates of benchmark prompts? Any safeguards for harmful or biased contexts when using LLMs to scaffold prompts?

---

> ### Author Response · Authors · 2025-11-22
> **Response to Reviewer tM5B (1/3)**
>
> We sincerely appreciate your positive assessment of our work and the time you devoted to reviewing our paper. Below, we provide detailed responses to each of your comments.
>
>
>
> ---
> **W1. Prompt Set Bias and its Dependence.**
> > *Dependence on curated prompt sets and LLMs: The large, hand/LLM-crafted prompt bank (tens of thousands) is central to performance. The construction process, coverage, and potential bias are only partially formalized; generalization to other domains/languages remains unclear.*
>
> * **Prompt set bias:** We acknowledge that a potential bias can exist in the constructed prompt set, and we examine this by conducting personalization with different sources of prompts. For this, we compared our template-based prompt result with the LLM-generated prompts and the human-written prompts. For human-written prompts, we adopted the COCO caption and replaced the corresponding subject word in each prompt with the personal concept tokens. Note that the prompt set used in our paper is template-based, constructed by combining contextual words from an LLM with templates, and differs from the LLM prompt used here.
> | **Prompt Source**     | **CLIP**        |
> |----------------|-------------|
> | Baseline | 0.770 |
> | COCO (human) | 0.788 |
> | LLM | 0.790 |
> | Template | 0.786 |
>
> The result shows that the model consistently improves over the baseline without MLM for all sources, and there is a small variation in CLIP score, indicating that our method is not overly sensitive to the specific choice of prompt set, reducing concerns about bias.
>
>
> * **Dependence on large prompts:** We would also like to note that constructing diverse text prompts is substantially more cost-effective than collecting images that depict a personal concept across diverse contexts (e.g., background, style). We note that the tens of thousands of prompts used in MLM training are not obtained by a costly large number of LLM inferences. Instead, we only query the LLM once to obtain each contextual vocabulary category, e.g., background, and the full prompt is constructed by combining these vocabularies with a small set of fixed template structures. As we will release the vocabulary and templates, this effort can be reduced further.
>
> * **Construction process:** As outlined in Appendix A.3, our prompt set is built through a template-driven procedure. This is achieved by first defining five context categories used during MLM training: human interactions, relative positions, backgrounds, image styles, and attribute variations. For each category, we prepare a small collection of template structures. We then query an LLM to obtain contextual vocabularies for that category and populate the templates with these context words. This produces a diverse set of prompts that describe the personal concept token across a wide range of contexts.
>
>
>
>
>
>
>
>
>
>
>
>
>
>
>
> ---
> **W2. The integration details of the MLM head.**
>
> > *Extra component and training stage: Requires pretraining a separate MLM head (albeit lightweight) on COCO+constructed prompts. The integration details (which layers are updated, interaction with CLIP tokenization nuances) could be more rigorously specified.*
>
>
> We provide additional technical details describing how the MLM head is integrated during 1) pretraining and 2) T2I personalization.
>
> * **Pretraining:** During pretraining, an MLM head consisting of four self-attention layers, each followed by a feed-forward layer, is appended to the CLIP text encoder. The input token embeddings of CLIP and the layers are kept frozen. During training, the input text tokens are replaced with a newly introduced mask token embedding. The layers are trained with a classification objective to predict the correct masked token, and the mask token embedding is jointly optimized.
>
> * **T2I personalization:** During T2I personalization, we introduce the LoRA parameter into the CLIP text encoder. The CLIP encoder, including the MLM head, remains frozen, and only the LoRA parameters are updated. We use a LoRA rank of 32. Except for the token that corresponds to the personalized concept, all other input token embeddings are kept frozen.

---

> ### Author Response · Authors · 2025-11-22
> **Response to Reviewer tM5B (2/3)**
>
> ---
> **W3. Are there alternative strategies to LoRA-based adaptation on a frozen CLIP encoder?**
>
> > *It also assumes MLM is beneficial while keeping CLIP frozen beyond LoRA, which may not always hold.*
>
>
> We examined different fine-tuning strategies by updating the top three layers of the CLIP text encoder and compared the results with the proposed LoRA with a frozen CLIP text encoder.
>
> | **Fine-tuning**   | **CLIP ↑** |
> |----------------|-------------|
> | Top-3 Layers (w/o MLM)|0.693 |
> | Top-3 Layers (w/ MLM) |**0.769** |
> | LoRA (w/o MLM)         |0.562 |
> | LoRA (w/ MLM)         |**0.790** |
>
> As shown above, the MLM improves performance for both settings. Updating only the top layers introduces a lower degree of drift in the pretrained encoder; however, the performance gain was also marginal compared to entire model training with LoRA, as LoRA affects the entire model.
>
>
>
> ---
> **W4. User study and failure analysis by prompt category.**
> > *Metric reliance: CLIP, DINO, ImageReward, BLIP-VQA have known limitations. There is no human user study or identity retrieval evaluation to triangulate improvements, and no failure analysis by prompt category (e.g., heavily compositional, relational, or negations).*
>
>
>
> * **User study:** We conducted a pair-wise human evaluation comparing our method with DisenBooth [1], DB [2], CD [3], and XTI [4]. We recruited 14 participants, and each participant was shown 10 prompts for each baseline. For every prompt, participants were shown the baseline result alongside ours and asked to select the image that shows better results in identity or the prompt alignment. We report the win, lose, and tie rates of our method against each baseline, where "Win" indicates that our method was preferred.
>
> | **Prompt Align**   | **Win (Ours)** | **Lose** | **Tie** |
> |----------------|-------------|--------|------|
> | vs DisenBooth [1] |     **43**      |   21   |  36  |
> | vs CD [2]         |     **47**      |   17   |  36  |
> | vs XTI [3]         |     **51**      |   14   |  35  |
> | vs DB [4]        |     **50**      |   11   |  39  |
>
> | **Identity Align**   | **Win (Ours)** | **Lose** | **Tie** |
> |----------------|-------------|--------|------|
> | vs DisenBooth [1] |     **35**      |   26   |  39  |
> | vs CD [2]         |     **36**      |   28   |  36  |
> | vs XTI [3]         |     **33**      |   20   |  47  |
> | vs DB [4]        |     **70**      |   8    |  22 |
>
> As shown in the table, our method clearly outperformed all the baselines in all aspects, further demonstrating its effectiveness.
>
> * **Failure analysis**: We revised the paper to include qualitative failure cases that analyze the compositional, relational, and negation prompts, presented in Figure C of the Appendix. The figure also compares our outputs with those of the non-personalized base T2I model, and the similar errors observed in both models suggest that these issues primarily arise from the base model.

---

> ### Author Response · Authors · 2025-11-22
> **Response to Reviewer tM5B (3/3)**
>
> ---
> **Q1. Can the proposed method handle multiple personal concepts?**
> > *Multiple personalized tokens and continual personalization: Can the approach handle multiple personal concepts jointly, or sequential personalization without interference?*
>
>
> We evaluate the approach in a multi-concept personalization setting and compare it against CD [2]:
> | **Baseline**       | **CLIP ↑**        | **DINO ↑** |
> |----------------|-------------|--------|
> | CD [2]            | 0.799       |0.628|
> | Ours           | **0.824**   |**0.685**|
>
> As shown, our method achieves superior CLIP and DINO scores, highlighting the effectiveness of our approach in prompt alignment and subject fidelity is not limited to single concept personalization. We also would like to note that collecting images depicting multiple concepts in diverse contexts can be even more costly than single-concept data. However, the proposed approach effectively circumvents this constraint by achieving diversification and learning in the textual space with no additional cost in achieving effective multi-concept personalization.
>
>
>
>
>
>
>
>
>
> ---
> **Q2. Is there any safeguard for leakage and safety?**
> > *Data leakage and safety: How do you ensure the MLM prompt set does not inadvertently include near-duplicates of benchmark prompts? Any safeguards for harmful or biased contexts when using LLMs to scaffold prompts?*
>
>
> Thank you for raising this important point regarding leakage and safety. As described in Section 3.1 in our paper, the MLM prompt set is constructed to strictly exclude the benchmark prompt. We would like to note that our prompt set is template-based, and the LLM is used only to provide contextual vocabularies that are later combined with fixed templates. Having an LLM generate full prompts could inadvertently produce near-duplicate benchmark prompts or introduce harmful phrasing, and detection of such prompts is difficult due to high variations in the sentence structures. In contrast, the risk can be easily minimized with the template-based setup by inspecting the contextual vocabularies before inserting them into the fixed templates. We note that contextual vocabulary was manually inspected to ensure undesirable or biased content is not introduced during training.
>
>
>
>
>
> [1] DisenBooth: Identity-Preserving Disentangled Tuning for Subject-Driven Text-to-Image Generation, ICLR 2024.
> [2] Multi-Concept Customization of Text-to-Image Diffusion, CVPR 2023.
> [3] P+: Extended Textual Conditioning in Text-to-Image Generation, Arxiv 2023.
> [4] DreamBooth: Fine Tuning Text-to-Image Diffusion Models for Subject-Driven Generation, CVPR 2023.

---

> ### Author Response · Authors · 2025-12-04
> **Additional Response to tM5B Regarding W2**
>
> We would like to provide additional information that may not have been fully addressed in our original response on the W2.
>
> **The necessity of MLM head Pretraining.**
> We first would like to clarify that the MLM head pretraining is only done **once** and is shared by all the concept personalization. We argue that the pretraining is necessary, as the MLM head with random initialization provides noisy signals during initial training steps. To validate this, we compare a pretrained MLM head with a randomly initialized head that is simultaneously trained during personalization:
>
> | **MLM Head**           | **ImageReward ↑** | **CLIP ↑** |
> |------------------  |------    |------|
> | Baseline    (w/o MLM)|0.562     |0.774 |
> | Random Init.(w/ MLM) |0.579     |0.780 |
> | Pretrain    (w/ MLM) |**0.781** | **0.787**|
>
> As shown, the random initialization still leads to improvements from baseline (+0.017 ImageReward); however, it exhibits a substantial gap compared to the head with pretraining (-0.202 ImageReward). This is primarily due to noisy signals provided during the initial training steps, which can be inaccurate. Since personalization is done with a small number of training steps, this noisy signal during the initial steps can be highly influential. Thus, a pretrained MLM head is needed to provide reliable guidance for personalization.

---

### Author Response · Authors · 2025-12-04
**Summary of Rebuttal (1/2)**

Dear Area Chair,

We sincerely appreciate the substantial effort involved in stepping in at this stage. To assist your evaluation, we summarize below how our responses have resolved the **concerns** raised by reviewers, along with the **contributions** acknowledged by them. The revisions in the manuscripts are also denoted in **red**.



## Summary of Concerns Raised by Reviewers ##
---

**1. Reviewer YPra - Score: 4 (Provided initial response, but no follow-up discussion)**
After we submitted our response, the reviewer noted that *"the overall assessment has improved"*, though 2 concerns remained. We provided a follow-up response, but due to the incident, we could not receive further feedback. We strongly believe our follow-up response has fully resolved the remaining concerns. The key points are summarized below,

* **Resolved Concerns:**
  * **W3. Comparison with more recent and stronger personalization baselines (resolved).** We provided additional comparison showing **our method outperforms the recent baselines in both prompt and subject alignment**.

  * **W4. Extension to fine-grained categories (resolved).** We validated that our method effectively improves the baseline on facial concept images and provided supporting qualitative examples in **Figure B** in the appendix.

* **Remaining Concerns:**
  * **W2. Additional interpretable evidence with validation.** We provided additional interpretable evidence in **Figure D** of **Section C.1** in the appendix, where we have visualized the distributions of text embeddings. It is clearly shows that the baseline without our approach leads to a highly concentrated distribution, whereas ours are dispersed. This difference is evident in the figure, and provides strong evidence of our key claim that **our method enhances textual representation**.

  * **W1. Novelty.** We clarified that our paper is not limited to a "conceptual re-formulation", but provides a novel contribution. We would like to note that existing works focused on addressing overfitting of models by **diversifying images**, while we introduce diversity in **text space**. Diversified text prompts cannot easily be integrated for training without paired images. We provided a novel and practical solution by **learning the contexts** also in **text space**, as acknowledged by **Reviewer SYkC and tM5B**.



---
**2. Reviewer tM5B - Score: 6 (No response to rebuttal)**
* **Concerns:**
  * **W1. Prompt set bias.** We validated that there are marginal differences in performance across different sources, showing that the results are not sensitive to the prompt sources.

  * **W2. MLM head.** We provided a detailed description of the MLM head, and further experiments are conducted to show a substantial decrease in performance when no pretraining is involved, which underscores its importance.

  * **W3. Ablation on the finetuning strategy.** We added additional ablations comparing different finetuning strategies, showing that the effectiveness is not limited to a single strategy.

  * **W4. Reliable metrics.** To validate with more reliable metrics, we provided a user study showing that our method **significantly outperforms all the baselines**.





---
**3. Reviewer SYkC - Score: 8 (No response to rebuttal)**
* **Concerns:**
  * **W1. Prompt set bias.** We conducted experiments using multiple types of prompt sets and confirmed that the results remain stable across them.

  * **Q1. Ablation with a linear layer on top of a partially fine-tuned CLIP encoder.** We added additional ablations on the suggested setting, and showed consistent improvements.





---
**4. Reviewer TZWq - Score: 2 (No response to rebuttal)**

* **Concerns:**
  * **W1. Tuning-based method vs learning-based method.** We provided a direct comparison with learning-based approaches and showed that our method **substantially outperforms** them. This result highlights the strength of our design and confirms that it is not outdated.

  * **W5. Comparison with more recent baselines.** We added additional experiments demonstrating that **our method outperforms recent baselines in both prompt and subject alignment.**

  * **W2. Why MLM for personalization?** We clarified that diversified text prompts **cannot** be directly integrated into personalization because they **require paired images**. The MLM serves as an effective way to overcome this limitation, as it operates entirely in text space.

  * **W3. How is the linear combination of input tokens related to personalization?** We clarified how the linear combination achieves contextualization of personal concept with diversified contexts in **text space**, the key contribution of our work.

  * **W4. Why does textual enhancement affect cross-attention?** We provided an additional mathematical explanation on **why** the enhancement in text space leads to better estimation of cross-attention. We have provided detailed descriptions in **Section C.2**. in the appendix.

---

> ### Author Response · Authors · 2025-12-04
> **Summary of Rebuttal (2/2)**
>
> ## Summary of Strengths Acknowledged by Reviewers ##
> ---
>
> **1. Novelty (Reviewers YPra, SYkC).** The key contribution of diversifying and learning the textual contexts has been acknowledged to be interesting and novel.
>
> **2. Strong performance (Reviewers tM5B, SYkC, TZWq).** Our method achieves strong empirical results on widely-used benchmarks with state-of-the-art results.
>
> **3. Cost-effective solution (Reviewers tM5B, YPra).** Unlike the existing approaches that address the overfitting of the models by collecting diverse images, we introduce diversity in **text space**, thereby offering a cost-effective solution.
>
> **4. Analysis and ablations (Reviewers tM5B, YPra).** We presented an analytical study showing how our approach influences representations in text space through cosine distance analysis of text embeddings, and how these effects propagate to image space via cross-attention maps analysis, providing evidence of its effectiveness.
>
> **5. Writing and Presentation (Reviewers tM5B, SYkC, TZWq).** The reviewers acknowledge the quality of the presentation and confirmed that it is easy to follow.
>
>
> Sincerely,
>
> Authors

---

### Note · Authors · 2026-01-28

**Comment:**

We thank the reviewers and ACs for their constructive and encouraging feedback, including their recognition of the framework’s strengths and their valuable suggestions for improvement.

**Withdrawal Confirmation:**

I have read and agree with the venue's withdrawal policy on behalf of myself and my co-authors.

---

### Meta-Review · Area_Chair_UWkV · 2025-12-21

**Summary:**

The reviewers’ discussion primarily focused on the limited methodological novelty of the proposed approach, despite its solid empirical performance. The paper reframes robustness issues in text-to-image personalization as a lack of contextual diversity and addresses this by introducing diverse textual prompts combined with an MLM objective during personalization. While this design is technically sound and well executed, several reviewers raised concerns that the core components—contextual prompt diversification, masked token prediction, and attention-based semantic regularization—have each been widely explored in related personalization and representation-learning literature, and their combination here represents an incremental extension rather than a fundamentally new paradigm.

Additional concerns included the reliance on manually or template-constructed prompt sets, the need for an auxiliary MLM training stage, and whether the observed gains reflect a principled advance beyond prompt engineering or regularization effects. Some reviewers also questioned the generality of the approach to more fine-grained identity settings and whether the method meaningfully differs from prior work that improves personalization by reducing semantic entanglement in text space.

In the rebuttal, the authors provided substantial clarifications, additional ablations, and stronger empirical comparisons, which helped address most technical questions regarding robustness, prompt-source bias, and the mechanism linking text-space diversity to improved cross-attention and image alignment. These responses strengthened confidence in the correctness and effectiveness of the method. However, they do not fully alleviate the concern that the contribution is primarily a careful integration and validation of existing ideas, rather than a novel algorithmic or conceptual advance.

Overall, the reviewers converged on the view that the paper demonstrates strong experimental results and a clear engineering contribution, but that its innovation is limited in scope, which significantly influenced the suggested decision.

**Reviewer Concerns:**

Addressed by the rebuttal:
(1) The authors clarified the mechanism by which textual diversity learned via MLM improves image-space alignment, supported by additional text-space analysis and cross-attention visualizations.
(2) Concerns about prompt-source bias and robustness were addressed through added experiments across different prompt construction strategies.
(3) Requests for stronger baselines, ablations, and clearer empirical justification were largely satisfied, strengthening confidence in the reported performance gains.

Outstanding concerns:
(1) The core concern regarding limited methodological novelty remains. The approach largely combines existing ideas (prompt diversification, MLM-style objectives, attention regularization) and represents an incremental improvement rather than a fundamentally new contribution.
(2) The method still relies on manually or template-designed prompts and an auxiliary MLM training stage, raising questions about scalability and generalization to more challenging personalization settings.

**Reviewer Scores:**

Reviewer TM5B: Likely a slight increase.
This reviewer raised several technical and empirical concerns (prompt source bias, MLM design choices, evaluation completeness). The rebuttal provided additional experiments, ablations, and clarifications that largely addressed these issues, improving confidence in the correctness and robustness of the approach. However, since novelty was not a primary strength to begin with, any increase would be modest.

Reviewer SYKC: Likely no change.
Although the rebuttal clarified how textual diversity affects attention and added supporting experiments, this reviewer’s main concerns centered on conceptual clarity and the incremental nature of the contribution. These concerns were only partially alleviated, suggesting the score would remain largely unchanged.

Reviewer YPra: Likely no change or at most a marginal increase.
The additional analysis and comparisons helped address questions about mechanism and empirical validation. However, the reviewer explicitly questioned whether the method goes beyond textual data augmentation and whether the novelty is substantial. As these concerns persist, any score increase would be minimal.

Reviewer TZWq: Likely no change.
This reviewer expressed skepticism about the learning-based framing and the necessity of MLM, viewing the approach as close to prompt-based heuristics. While the rebuttal improved clarity and added comparisons, it did not fundamentally alter the reviewer’s assessment of limited innovation.

---

### Decision · Program_Chairs · 2026-01-26

Reject